# Efficient Training of
# Low-Curvature Neural Networks

**Suraj Srinivas**[*1]
Harvard University
ssrinivas@seas.harvard.edu

**Kyle Matoba**[*]
Idiap Research Institute & EPFL
kyle.matoba@epfl.ch

**Himabindu Lakkaraju**
Harvard University
hlakkaraju@hbs.edu

**François Fleuret**
University of Geneva
francois.fleuret@unige.ch

## Abstract

Standard deep neural networks often have excess non-linearity, making them susceptible to issues such as low adversarial robustness and gradient instability. Common methods to address these downstream issues, such as adversarial training, are expensive and often sacrifice predictive accuracy.

In this work, we address the core issue of excess non-linearity via curvature, and demonstrate low-curvature neural networks (LCNNs) that obtain drastically lower curvature than standard models while exhibiting similar predictive performance. This leads to improved robustness and stable gradients, at a fraction of the cost of standard adversarial training. To achieve this, we decompose overall model curvature in terms of curvatures and slopes of its constituent layers. To enable efficient curvature minimization of constituent layers, we introduce two novel architectural components: first, a non-linearity called centered-softplus that is a stable variant of the softplus non-linearity, and second, a Lipschitz-constrained batch normalization layer.

Our experiments show that LCNNs have lower curvature, more stable gradients and increased off-the-shelf adversarial robustness when compared to standard neural networks, all without affecting predictive performance. Our approach is easy to use and can be readily incorporated into existing neural network architectures.

Code to implement our method and replicate our experiments is available at https://github.com/kylematoba/lcnn.

## 1 Introduction

The high degree of flexibility present in deep neural networks is critical to achieving good performance in complex tasks such as image classification, language modelling and generative modelling of images [1–3]. However, *excessive* flexibility is undesirable as this can lead to model under-specification [4] which results in unpredictable behaviour on out-of-domain inputs, such as vulnerability to adversarial examples. Such under-specification can be avoided in principle via Occam's razor, which requires training models that are as simple as possible for the task at hand, and not any more. Motivated by this principle, in this work we focus on training neural network models without excess non-linearity in their input-output map (e.g.: see Fig 1b), such that predictive performance remains unaffected.

---

[*]Equal Contribution
[1]Work done partially at Idiap Research Institute

36th Conference on Neural Information Processing Systems (NeurIPS 2022).

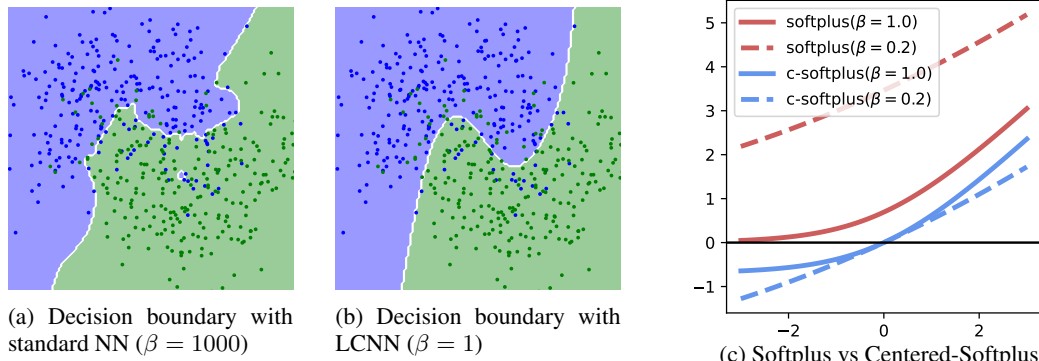

(a) Decision boundary with standard NN ($\beta = 1000$)

(b) Decision boundary with LCNN ($\beta = 1$)

(c) Softplus vs Centered-Softplus

Figure 1: Decision boundaries of (a) standard NN and (b) LCNN trained on the two moons dataset. LCNN recovers highly regular decision boundaries in contrast to the standard NN. (c) Comparison of softplus and centered-softplus non-linearities (defined in §3.3). These behave similarly for large $\beta$ values, and converge to linear maps for low $\beta$ values. However, softplus diverges while centered-softplus stays close to the origin.

Central to this work is a precise notion of curvature, which is a mathematical quantity that encodes the flexibility or the degree of non-linearity of a function at a point. In deep learning, the curvature of a function at a point is often quantified as the norm of the Hessian at that point [5–7]. Hessian norms are zero everywhere if and only if the function is linear, making them suitable to measure the degree of non-linearity. However, they suffer from a dependence on the scaling of model gradients, which makes them unsuitable to study its interplay with model robustness. In particular, robust models have small gradient norms [8], which naturally imply smaller Hessian norms. But are they truly more linear as a result? To be able to study robustness independent of non-linearity, we propose *normalized curvature*, which normalizes the Hessian norm by its corresponding gradient norm, thus disentangling the two measures. Surprisingly, we find that normalized curvature is a stable measure across train and test samples (see Table 3), whereas the usual curvature is not.

A conceptually straightforward approach to train models with low-curvature input-output maps [6, 9] involves directly penalizing curvature locally at training samples. However, these methods involve expensive Hessian computations, and only minimize local point-wise curvature and not curvature everywhere. A complementary approach is that of Dombrowski et al. [10], who propose architectures that have small global curvature, but do not explicitly penalize this curvature during training. In contrast, we propose efficient mechanisms to explicitly penalize the *normalized curvature* globally. In addition, while previous methods [6, 7] penalize the Frobenius norm of the Hessian, we penalize its spectral norm, thus providing tighter and more interpretable robustness bounds.[1]

Our overall contributions in this paper are:

1. In §3.1 we propose to measure curvature of deep models via *normalized curvature*, which is invariant to scaling of the model gradients.

2. In §3.2 we show that normalized curvature of neural networks can be upper bounded in terms of normalized curvatures and slopes of individual layers.

3. We introduce an architecture for training LCNNs combining a novel activation function called the centered-softplus (§3.3), and recent innovations to constrain the Lipschitz constant of convolutional (§3.4.1) and batch normalization (§3.4.2) layers.

4. In §4 we prove bounds on the relative gradient robustness and adversarial robustness of models in terms of normalized curvature, showing that controlling normalized curvature directly controls these properties.

5. In §5, we show experiments demonstrating that our proposed innovations are successful in training low-curvature models without sacrificing training accuracy, and that such models have robust gradients, and are more robust to adversarial examples out-of-the-box.

---

[1]Note that the Frobenius norm of a matrix strictly upper bounds its spectral norm

## 2    Related Work

**Adversarial Robustness of Neural Networks:** The well-known phenemenon of adversarial vulnerabilities of neural networks [11, 12] shows that adding small amounts of imperceptible noise can cause deep neural networks to misclassify points with high confidence. The canonical method to defend against this vulnerability is adversarial training [13] which trains models to accurately classify adversarial examples generated via an attack such as projected gradient descent (PGD). However, this approach is computationally expensive and provides no formal guarantees on robustness. Cohen et al. [14] proposed randomized smoothing, which provides a formal guarantee on robustness by generating a smooth classifer from any black-box classifier. Hein and Andriushchenko [8] identified the local Lipschitz constant as critical quantity to prove formal robustness guarantees. Moosavi Dezfooli et al. [6] penalize the Frobenius norm of the Hessian, and show that they performs similarly to models trained via adversarial training. Qin et al. [9] introduce a local linearity regularizer, which also implicitly penalizes the Hessian. Similar to these works, we enforce low curvature to induce robustness, but we focus on out of the box robustness of LCNNs.

**Unreliable Gradient Interpretations in Neural Networks:** Gradient explanations in neural networks can be unreliable. Ghorbani et al. [15], Zhang et al. [16] showed that for any input, it is possible to find adversarial inputs such that the gradient explanations for these points that are highly dissimilar to each other. Srinivas and Fleuret [17] showed that pre-softmax logit gradients however are independent of model behaviour, and as a result we focus on post-softmax loss gradients in this work. Ros and Doshi-Velez [18] showed empirically that robustness can be improved by gradient regularization, however Dombrowski et al. [10] showed that gradient instability is primarily due to large Hessian norms. This suggests that the gradient penalization in Ros and Doshi-Velez [18] performed unintentional Hessian regularization, which is consistent with our experimental results. To alleviate this, Dombrowski et al. [7] proposed to train low curvature models via softplus activations and weight decay, broadly similar to our strategy. However, while Dombrowski et al. [7] focused on the Frobenius norm of the Hessian, we penalize the normalized curvature – a scaled version of the Hessian spectral norm – which is strictly smaller than the Frobenius norm, and this results in a more sophisticated penalization strategy.

**Lipschitz Layers in Neural Networks** There has been extensive work on methods to bound the Lipschitz constant of neural networks. Cisse et al. [19] introduced Parseval networks, which penalizes the deviation of linear layers from orthonormality – since an orthonormal linear operator evidently has a Lipschitz constant of one, this shrinks the Lipschitz constant of a layer towards one. Trockman and Kolter [20] use a reparameterization of the weight matrix, called the Cayley Transform, that is orthogonal by construction. Miyato et al. [21], Ryu et al. [22] proposed spectral normalization, where linear layers are re-parameterized by dividing by their spectral norm, ensuring that the overall spectral norm of the parameterized layer is one.

## 3    Training Low-Curvature Neural Networks

In this section, we introduce our approach for training low-curvature neural nets (LCNNs). Unless otherwise specified, we shall consider a neural network classifier $f$ that maps inputs $\mathbf{x} \in \mathbb{R}^d$ to logits which characterize the prediction, and can be further combined with the true label distribution and a loss function to give a scalar loss value $f(\mathbf{x}) \in \mathbb{R}^+$.

### 3.1    Measuring Relative Model Non-Linearity via Normalized Curvature

We begin our analysis by discussing a desirable definition of curvature $\mathcal{C}_f(\mathbf{x}) \in \mathbb{R}^+$. While curvature is well-studied topic in differential geometry [23] where Hessian normalization is a common theme, our discussion will be largely independent of this literature, in a manner more suited to requirements in deep learning. Regardless of the definition, a typical property of a curvature measure is that $\mathcal{C}_f(\mathbf{x}) = 0 \ \forall \mathbf{x} \in \mathbb{R}^d \iff f$ is linear, and the higher the curvature, the further from linear the function is. Hence $\max_{\mathbf{x} \in \mathbb{R}^d} \mathcal{C}_f(\mathbf{x})$ can be thought of as a measure of a model's non-linearity. A common way to define curvature in machine learning [5–7] has been via Hessian norms. However, these measures are sensitive to gradient scaling, which is undesirable. After all, the degree of model non-linearity must intuitively be independent of how large its gradients are.

For example, if two functions $f, g$ are scaled ($f = k \times g$), or rotated versions of each other ($\nabla f = k \times \nabla g$), then we would like them to have similar curvatures in the interest of disentangling curvature (i.e., degree of non-linearity) from scaling. It is easy to see that Hessian norms $\|\nabla_{\mathbf{x}}^2 f(\mathbf{x})\|_2$ do not have this property, as scaling the function also scales the corresponding Hessian. We would like to be able to recover low-curvature models with steep gradients (that are non-robust), precisely to be able to disentangle their properties from low-curvature models with small gradients (that are robust), which Hessian norms do not allow us to do.

To avoid this problem, we propose a definition of curvature that is approximately *normalized*: $\mathcal{C}_f(\mathbf{x}) = \|\nabla^2 f(\mathbf{x})\|_2 / (\|\nabla f(\mathbf{x})\|_2 + \varepsilon)$. Here $\|\nabla f(\mathbf{x})\|_2$ and $\|\nabla^2 f(\mathbf{x})\|_2$ are the $\ell_2$ norm of the gradient and the spectral norm of the Hessian, respectively, where $\nabla f(\mathbf{x}) \in \mathbb{R}^d, \nabla^2 f(\mathbf{x}) \in \mathbb{R}^{d \times d}$, and $\varepsilon > 0$ is a small constant to ensure well-behavedness of the measure. This definition measures Hessian norm *relative* to the gradient norm, and captures a notion of relative local linearity, which can be seen via an application of Taylor's theorem:

$$\underbrace{\frac{\|f(\mathbf{x} + \epsilon) - f(\mathbf{x}) - \nabla f(\mathbf{x})^\top \epsilon\|_2}{\|\nabla f(\mathbf{x})\|_2}}_{\text{relative local linearity}} \leq \frac{1}{2} \underbrace{\max_{\mathbf{x} \in \mathbb{R}^d} \mathcal{C}_f(\mathbf{x})}_{\text{normalized curvature}} \|\epsilon\|_2. \tag{1}$$

Here the numerator on the left captures the local linearity error, scaled by the gradient norm in the denominator, which can be shown using Taylor's theorem to be upper bounded by the normalized curvature. We shall consider penalizing this notion of normalized curvature, and we shall simply refer to this quantity as curvature in the rest of the paper.

## 3.2 A Data-Free Upper Bound on Curvature

Directly penalizing the curvature is computationally expensive as it requires backpropagating an estimate of the Hessian norm, which itself requires backpropagating gradient-vector products. This requires chaining the entire computational graph of the model at least three times. Moosavi Dezfooli et al. [6] reduce the complexity of this operation by computing a finite-difference approximation to the Hessian from gradients, but even this double-backpropagation is expensive. We require an efficient penalization procedure takes a single backpropagation step.

To this end, we propose to minimize a data-free upper bound on the curvature. To illustrate the idea, we first show this upper bound for the simplified case of the composition of one-dimensional functions ($f : \mathbb{R} \to \mathbb{R}$).

**Lemma 1.** *Given a 1-dimensional compositional function $f = f_L \circ f_{L-1} \circ \ldots \circ f_1$ with $f_i : \mathbb{R} \to \mathbb{R}$ for $i = 1, 2, \ldots, L$, the normalized curvature $\mathcal{C}_f := |f''/f'|$, is bounded by $\left|\frac{f''}{f'}\right| \leq \sum_{i=1}^{L} \left|\frac{f_i''}{f_i'}\right| \prod_{j=1}^{i} |f_j'|$*

*Proof.* We first have $f' = \prod_{i=1}^{L} f_i'$. Differentiating this expression we have $f'' = \sum_{i=1}^{L} f_i'' \prod_{j=1}^{i} f_j' \prod_{k=1, k \neq i}^{L} f_k'$. Dividing by $f'$, taking absolute value of both sides, and using the triangle inequality, we have the intended result.

Extending the above expression to functions $\mathbb{R}^n \to \mathbb{R}$ is not straightforward, as the intermediate layer Hessians are order three tensors. We derive this using a result from Wang et al. [24] that connects the spectral norms of order-$n$ tensors to the spectral norm of their matrix "unfoldings". The full derivation is presented in the appendix, and the (simplified) result is stated below.

**Theorem 1.** *Given a function $f = f_L \circ f_{L-1} \circ \ldots \circ f_1$ with $f_i : \mathbb{R}^{n_{i-1}} \to \mathbb{R}^{n_i}$, the curvature $\mathcal{C}_f$ can be bounded by the sum of curvatures of individual layers $\mathcal{C}_{f_i}(\mathbf{x})$, i.e.,*

$$\mathcal{C}_f(\mathbf{x}) \leq \sum_{i=1}^{L} n_i \times \mathcal{C}_{f_i}(\mathbf{x}) \prod_{j=1}^{i} \|\nabla_{f_{j-1}} f_j(\mathbf{x})\|_2 \leq \sum_{i=1}^{L} n_i \times \max_{\mathbf{x}'} \mathcal{C}_{f_i}(\mathbf{x}') \prod_{j=1}^{i} \max_{\mathbf{x}'} \|\nabla_{f_{j-1}} f_j(\mathbf{x}')\|_2.$$
$$\tag{2}$$

The rightmost term is independent of $\mathbf{x}$ and thus holds uniformly across all data points. This bound shows that controlling the curvature and Lipschitz constant of each layer of a neural network enables us to control the overall curvature of the model.

Practical neural networks typically consist of[1] linear maps such as convolutions, fully connected layers, and batch normalization layers, and non-linear activation functions. Linear maps have zero curvature by definition, and non-linear layers often have bounded gradients ($= 1$), which simplifies computations. In the sections that follow, we shall see how to penalize the remaining terms, i.e., the curvature of the non-linear activations, and the Lipschitz constant of the linear layers.

### 3.3 Centered-Softplus: Activation Function with Trainable Curvature

Theorem 1 shows that the curvature of a neural network depends on the curvature of its constituent activation functions. We thus propose to use activation functions with minimal curvature.

The kink at the origin of the ReLU function implies an undefined second derivative. On the other hand, a smooth activation such as the softplus function, $s(x; \beta) = \log(1 + \exp(\beta x))/\beta$ is better suited to analyzing questions of curvature. Despite not being a common baseline choice, softplus does see regular use, especially where its smoothness facilitates analysis [25]. The curvature of the softplus function is

$$\mathcal{C}_{s(\cdot;\beta)}(x) = \beta \times \left(1 - \frac{\mathrm{d}s(x;\beta)}{\mathrm{d}x}\right) \leq \beta \tag{3}$$

Thus using softplus with small $\beta$ values ensures low curvature. However, we observe two critical drawbacks of softplus preventing its usage with small $\beta$: (1) divergence for small $\beta$, where $s(x; \beta \to 0) = \infty$ which ensures that well-behaved low curvature maps cannot be recovered and, (2) instability upon composition i.e., $s^n(x = 0; \beta) = \underbrace{s \circ s \circ \dots \circ s}_{n \text{ times}}(x = 0; \beta) = \frac{\log(n+1)}{\beta}$. Thus we have that $s^n(x = 0; \beta) \to \infty$ as $n \to \infty$, which shows that composing softplus functions exacerbates instability around the origin. This is critical for deep models with large number of layers $n$, which is precisely the scenario of interest to us. To remedy this problem, we propose *centered-softplus* $s_0(x; \beta)$, a simple modification to softplus by introducing a normalizing term as follows.

$$s_0(x; \beta) = s(x; \beta) - \frac{\log 2}{\beta} = \frac{1}{\beta} \log\left(\frac{1 + \exp(\beta x)}{2}\right). \tag{4}$$

This ensures that $s(x = 0; \beta) = s^n(x = 0; \beta) = 0$ for any positive integer $n$, and hence also ensures stability upon composition. More importantly, we have $s_0(x; \beta \to 0) = x/2$ which is a scaled linear map, while still retaining $s_0(x; \beta \to \infty) = \text{ReLU}(x)$. This ensures that we are able to learn both well-behaved linear maps, as well as highly non-linear ReLU-like maps if required.

We further propose to cast $\beta$ is a learnable parameter and penalize its value, hence directly penalizing the curvature of that layer. Having accounted for the curvature of the non-linearities, the next section discusses controlling the gradients of the linear layers.

### 3.4 Lipschitz Linear Layers

Theorem 1 shows that penalizing the Lipschitz constants of the constituent linear layers in a model is necessary to penalize overall model curvature. There are broadly three classes of linear layers we consider: convolutions, fully connected layers, and batch normalization.

#### 3.4.1 Spectrally Normalized Convolutions and Fully Connected Layers

We propose to penalize the Lipschitz constant of convolutions and fully connected layer via existing spectral normalization-like techniques. For fully connected layers, we use vanilla spectral normalization (Miyato et al. [21]) which controls the spectral norm of a fully connected layer by reparameterization – replacing a weight matrix $W$, by $W/||W||_2$ which has a unit spectral norm. Ryu et al. [22] generalize this to convolutions by presenting a power iteration method that works directly on the linear mapping implicit in a convolution: maintaining 3D left and right singular "vectors" of the 4D tensor of convolutional filters and developing the corresponding update rules.

---

[1]We ignore self-attention layers in this work.

They call this "real" spectral normalization to distinguish it from the approximation that [21] proposed for convolutions. Using spectral normalization on fully connected layers and "real" spectral normalization on convolutional layers ensures that the spectral norm of these layers is exactly equal to one, further simplifying the bound in Theorem 1.

### 3.4.2  $\gamma$-Lipschitz Batch Normalization

In principle, at inference time batch normalization (BN) is multiplication by the diagonal matrix of inverse running standard deviation estimates.[1] Thus we can spectrally normalize the BN layer by computing the reciprocal of the smallest running standard deviation value across dimensions $i$, i.e., $||\text{BN}||_2 = \max_x ||\text{BN}(x)||_2 = 1/\min_i \text{running-std}(i)$. In practice, we found that models with such spectrally normalized BN layers tended to either diverge or fail to train in the first place, indicating that the scale introduced by BN is necessary for training. To remedy this, we introduce the $\gamma$-*Lipschitz* batch normalization, defined by

$$\text{1-Lipschitz-BN}(\mathbf{x}) \leftarrow \text{BN}(\mathbf{x})/||\text{BN}||_2$$
$$\gamma\text{-Lipschitz-BN}(\mathbf{x}) \leftarrow \underbrace{\min(\gamma, ||\text{BN}||_2)}_{\text{scaling factor} \leq \gamma} \times \text{1-Lipschitz-BN}(\mathbf{x}).$$

By clipping the scaling above at $\gamma$ (equivalently, the running standard deviation below, at $1/\gamma$), we can ensure that the Lipschitz constant of a batch normalization layer is at most equal to $\gamma \in \mathbb{R}^+$. We provide a PyTorch-style code snippet in the appendix. As with $\beta$, described in §3.3, we cast $\gamma$ as a learnable parameter in order to penalize it during training. Gouk et al. [26] proposed a similar (simpler) solution, whereas they they fit a common $\gamma$ to all BN layers, we let $\gamma$ vary freely by layer.

### 3.5  Penalizing Curvature

We have discussed three architectural innovations – centered-softplus activations with a trainable $\beta$, spectrally normalized linear and convolution layers and a $\gamma$-Lipschitz batch normalization layer. We now discuss methods to penalize the overall curvature of a model built with these layers.

Since convolutional and fully connected layers have spectral norms equal to one by construction, they contribute nothing to curvature. Thus, we restrict attention to batch normalization and activation layers. The set of which will be subsequently referred to respectively $\beta$SP and $\gamma$BN. For models with batch normalization, naively using the upper bound in Theorem 1 is problematic due to the exponential growth in the product of Lipschitz constants of batch normalization layers. To alleviate this, we propose to use a penalization $\mathcal{R}_f$ where $\gamma_i$ terms are aggregated additively across batch normalization layers, and independent of the $\beta_i$ terms in the following manner:

$$\mathcal{R}_f = \lambda_\beta \times \sum_{i \in \beta\text{SP}} \beta_i + \lambda_\gamma \times \sum_{j \in \gamma\text{BN}} \log \gamma_j \tag{5}$$

An additive aggregation ensures that the penalization is well-behaved during training and does not grow exponentially. Note that the underlying model is necessarily linear if the penalization term is zero, thus making it an appropriate measure of model non-linearity. Also note that $\beta_i \geq 0, \gamma_i \geq 1$ by construction. We shall henceforth use the term "LCNN" to refer to a model trained with the proposed architectural components (centered-softplus, spectral normalization, and $\gamma$-Lipschitz batch normalization) and associated regularization terms on $\beta, \gamma$. We next discuss the robustness and interpretability benefits that we obtain with LCNNs.

## 4  Why Train Low-Curvature Models?

In this section we discuss the advantages that low-curvature models offer, particularly as it pertains to robustness and gradient stability. These statements apply not just to LCNNs, but low-curvature models in general obtained via any other mechanism.

---

[1] We ignore the learnable parameters of batch normalization for simplicity. The architecture we propose subsequently also does not have trainable affine parameters.

### 4.1 Low Curvature Models have Stable Gradients

Recent work [15, 16] has shown that gradient explanations are manipulable, and that we can easily find inputs whose explanations differ maximally from those at the original inputs, making them unreliable in practice for identifying important features. As we shall show, this amounts to models having a large curvature. In particular, we show that the relative gradient difference across points $\|\epsilon\|_2$ away in the $\ell_2$ sense is upper bounded by the normalized curvature $\mathcal{C}_f$, as given below.

**Proposition 1.** *Consider a model $f$ with $\max_{\mathbf{x}} \mathcal{C}_f(\mathbf{x}) \leq \delta_{\mathcal{C}}$, and two inputs $\mathbf{x}$ and $\mathbf{x} + \epsilon$ ($\in \mathbb{R}^d$). The relative distance between gradients at these points is bounded by*

$$\frac{\|\nabla f(\mathbf{x} + \epsilon) - \nabla f(\mathbf{x})\|_2}{\|\nabla f(\mathbf{x})\|_2} \leq \|\epsilon\|_2 \delta_{\mathcal{C}} \exp(\|\epsilon\|_2 \delta_{\mathcal{C}}) \sim \|\epsilon\|_2 \mathcal{C}_f(\mathbf{x}) \quad \textit{(Quadratic Approximation)}$$

The proof expands $f(\mathbf{x})$ in a Taylor expansion around $\mathbf{x} + \epsilon$ and bounds the magnitude of the second and higher order terms over the neighborhood of $\mathbf{x}$, and the full argument is given in the appendix. The upper bound is considerably simpler when we assume that the function locally quadratic, which corresponds to the rightmost term $\|\epsilon\|_2 \mathcal{C}_f(\mathbf{x})$. Thus the smaller the model curvature, the more locally stable are the gradients.

### 4.2 Low Curvature is Necessary for $\ell_2$ Robustness

Having a small gradient norm is known to be an important aspect of adversarial robustness [8]. However, small gradients alone are not sufficient, and low curvature is necessary, to achieve robustness. This is easy to see intuitively - a model may have low gradients at a point leading to robustness for small noise values, but if the curvature is large, then gradient norms at neighboring points can quickly increase, leading to misclassification for even slightly larger noise levels. This effect is an instance of *gradient-masking* [27], which provides an illusion of robustness by making models only locally robust.

In the result below, we formalize this intuition and establish an upper bound on the distance between two nearby points, which we show depends on both the gradient norm (as was known previously) and well as the max curvature of the underlying model.

**Proposition 2.** *Consider a model $f$ with $\max_{\mathbf{x}} \mathcal{C}_f(\mathbf{x}) \leq \delta_{\mathcal{C}}$, then for two inputs $\mathbf{x}$ and $\mathbf{x} + \epsilon$ ($\in \mathbb{R}^d$), we have the following expression for robustness*

$$\begin{aligned} \|f(\mathbf{x} + \epsilon) - f(\mathbf{x})\|_2 &\leq \|\epsilon\|_2 \|\nabla f(\mathbf{x})\|_2 \left(1 + \|\epsilon\|_2 \delta_{\mathcal{C}} \exp(\|\epsilon\|_2 \delta_{\mathcal{C}})\right) \\ &\sim \|\epsilon\|_2 \|\nabla f(\mathbf{x})\|_2 \left(1 + \|\epsilon\|_2 \mathcal{C}_f(\mathbf{x})\right) \quad \textit{(Quadratic Approximation)} \end{aligned}$$

The proof uses similar techniques to those of proposition 1, and is also given in the appendix. This result shows that given two models with equally small gradients at data points, the greater robustness will be achieved by the model with the smaller curvature.

## 5 Experiments

In this section we perform experiments to (1) evaluate the effectiveness of our proposed method in training models with low curvature as originally intended, (2) evaluate whether low curvature models have robust gradients in practice, and (3) evaluate the effectiveness of low-curvature models for adversarial robustness. Our experiments are primarily conducted on a base ResNet-18 architecture ([28]) using the CIFAR10 and CIFAR100 datasets ([29]), and using the Pytorch [30] framework. Our methods entailed fairly modest computation – our most involved computations can be completed in under three GPU days, and all experimental results could be computed in less than 60 GPU-days. We used a mixture of GPUs – primarily NVIDIA GeForce GTX 1080 Tis – on an internal compute cluster.

**Baselines and Our Methods** Our baseline model for comparison involves using a ResNet-18 model with softplus activation with a high $\beta = 10^3$ to mimic ReLU, and yet have well-defined curvatures.

Another baseline is gradient norm regularization (which we henceforth call 'GradReg') [31], where the same baseline model is trained with an additional penalty on the gradient norm. We train two variants of our approach - a base LCNN, which involves penalizing the curvature, and another variant combining LCNN penalty and gradient norm regularization (LCNN + GradReg), which controls both the curvature and gradient norm. Our theory indicates that the LCNN + GradReg variant is likely to produce more robust models, which we verify experimentally. We also compare with CURE [6], softplus with weight decay [7] and adversarial training with $\ell_2$ PGD [13] with noise magnitude of 0.1 and 3 iterations of PGD. We provide experimental details in the appendix.

**Parameter Settings** All our models are trained for 200 epochs with an SGD + momentum optimizer, with a momentum of 0.9 and an initial learning rate of 0.1 which decays by a factor of 10 at 150 and 175 epochs, and a weight decay of $5 \times 10^{-4}$.

## 5.1 Evaluating the Efficacy of Curvature Penalization

In this section, we evaluate whether LCNNs indeed reduce model curvature in practice. Table 1 contains our results, from which we make the following observations: (1) most of our baselines except CURE and adversarial training do not meaningfully lose predictive performance (2) GradReg and adversarially trained models are best at reducing gradient norm while LCNN-based models are best at penalizing curvature. Overall, these experimental results show that LCNN-based models indeed minimize curvature as intended.

We also observe in Table 1 that GradReg [31] has an unexpected regularizing effect on the Hessian and curvature. We conjecture that this is partly due to the following decomposition of the loss Hessian, which can be written as $\nabla^2 f(\mathbf{x}) \sim \nabla f_l(\mathbf{x}) \nabla_{f_l}^2 \mathrm{LSE}(\mathbf{x}) \nabla f_l(\mathbf{x})^\top + \nabla_{f_l} \mathrm{LSE}(\mathbf{x}) \nabla^2 f_l(\mathbf{x})$, where $\mathrm{LSE}(\mathbf{x})$ is the LogSumExp function, and $f_l(\mathbf{x})$ is the pre-softmax logit output. We observe that the first term strongly depends on the gradients, which may explain the Hessian regularization effects of GradReg, while the second term depends on the Hessian of the bulk of the neural network, which is penalized by the LCNN penalties. This also explains why combining both penalizations (LCNN + GradReg) further reduces curvature.

We also measure the average per-epoch training times on a GTX 1080Ti, which are: standard models / softplus + weight decay ($\sim 100$ sec), LCNN ($\sim 160$ sec), GradReg ($\sim 270$ sec), LCNN+GradReg ($\sim 350$ sec), CURE / Adversarial Training ($\sim 500$ sec). Note that the increase in computation for LCNN is primarily due to the use of spectral normalization layers. The results show that LCNNs are indeed able to penalize curvature by only marginally ($1.6\times$) increasing training time, and using LCNN+GradReg only increases time $1.3\times$ over GradReg while providing curvature benefits.

Table 1: Model geometry of various ResNet-18 models trained with various regularizers on the CIFAR100 test dataset. Gradient norm regularized models [31] ('GradReg') are best at reducing gradient norms, while LCNN-based models are best at reducing curvature, leaving gradients unpenalized. We obtain the benefits of both by combining these penalties. Results are averaged across two runs.

| Model | $\mathbb{E}_\mathbf{x}\|\nabla f(\mathbf{x})\|_2$ | $\mathbb{E}_\mathbf{x}\|\nabla^2 f(\mathbf{x})\|_2$ | $\mathbb{E}_\mathbf{x}\mathcal{C}_f(\mathbf{x})$ | Accuracy (%) |
|---|---|---|---|---|
| Standard | $19.66_{\pm 0.33}$ | $6061.96_{\pm 968.05}$ | $270.89_{\pm 75.04}$ | $77.42_{\pm 0.11}$ |
| LCNNs | $22.04_{\pm 1.41}$ | $1143.62_{\pm 99.38}$ | $69.50_{\pm 2.41}$ | $77.30_{\pm 0.11}$ |
| GradReg [31] | $\mathbf{8.86}_{\pm 0.12}$ | $776.56_{\pm 63.62}$ | $89.47_{\pm 5.86}$ | $77.20_{\pm 0.26}$ |
| LCNNs + GradReg | $9.87_{\pm 0.27}$ | $\mathbf{154.36}_{\pm 0.22}$ | $\mathbf{25.30}_{\pm 0.09}$ | $77.29_{\pm 0.07}$ |
| CURE [6] | $\mathbf{8.86}_{\pm 0.01}$ | $979.45_{\pm 14.05}$ | $116.31_{\pm 4.58}$ | $76.48_{\pm 0.07}$ |
| Softplus + Wt. Decay [7] | $18.08_{\pm 0.05}$ | $1052.84_{\pm 7.27}$ | $70.39_{\pm 0.88}$ | $77.44_{\pm 0.28}$ |
| Adversarial Training [32] | $\mathbf{7.99}_{\pm 0.03}$ | $501.43_{\pm 18.64}$ | $63.79_{\pm 1.65}$ | $76.96_{\pm 0.26}$ |

## 5.2 Impact of Curvature on Gradient Robustness

In §4.1, we showed that low-curvature models tend to have more robust gradients. Here we evaluate whether this prediction empirically by measuring the relative gradient robustness for the models with various ranges of curvature values and noise levels. In particular, we measure robustness to random noise at fixed magnitudes ranging logarithmically from $1 \times 10^{-3}$ to $1 \times 10^{-1}$. We plot our results

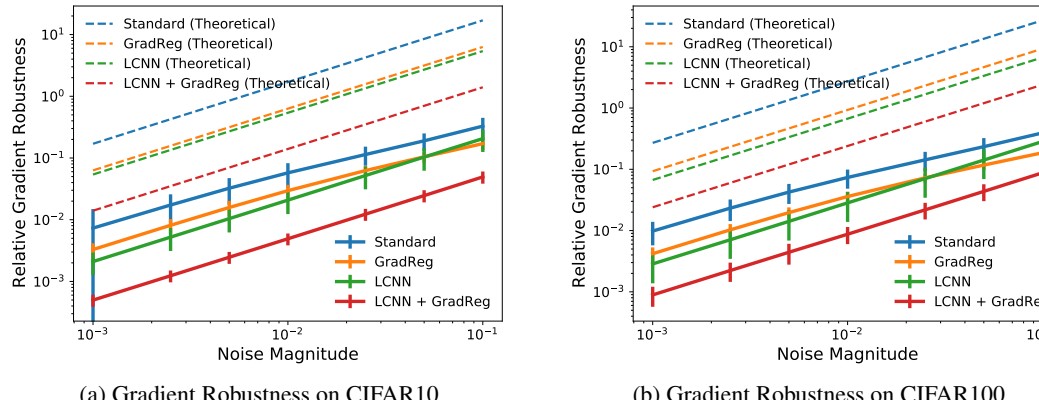

(a) Gradient Robustness on CIFAR10    (b) Gradient Robustness on CIFAR100

Figure 2: Plot showing relative gradient robustness $\frac{\|\nabla f(\mathbf{x}+\epsilon) - \nabla f(\mathbf{x})\|_2}{\|\nabla f(\mathbf{x})\|_2}$ as a function of added noise $\|\epsilon\|_2$ on (a) CIFAR10 and (b) CIFAR100 with a ResNet-18 model. We observe that low-curvature models lead to an **order of magnitude** improvement in gradient robustness, and this improvement closely follows the trend predicted by the theoretical upper bound in §4.

in Figure 2 where we find that our results match theory (i.e, the simplified quadratic approximation in §4.1) quite closely in terms of the overall trends, and that low curvature models have an **order of magnitude** improvement in robustness over standard models.

## 5.3    Impact of Curvature on Adversarial Robustness

Our theory in §4.2 shows that having low curvature is necessary for robustness, along with having small gradient norms. In this section we evaluate this claim empirically, by evaluating adversarial examples via $\ell_2$ PGD [13] adversaries with various noise magnitudes. We use the Cleverhans library [33] to implement PGD. Our results are present in Table 2 where we find that LCNN+GradReg models perform on par with adversarial training, without a resulting accuracy loss.

Table 2: Results indicating off-the-shelf model accuracies (%) upon using $\ell_2$ PGD adversarial examples across various noise magnitudes. Adversarial training performs the best overall, however sacrifices clean accuracy. LCNN+GradReg models perform similarly but without significant loss of clean accuracy. Results are averaged across two runs.

| **Model** | Acc. (%) | $\|\epsilon\|_2 = 0.05$ | $\|\epsilon\|_2 = 0.1$ | $\|\epsilon\|_2 = 0.15$ | $\|\epsilon\|_2 = 0.2$ |
|---|---|---|---|---|---|
| Standard | $77.42 \pm .10$ | $59.97 \pm .11$ | $37.55 \pm .13$ | $23.41 \pm .08$ | $16.11 \pm .21$ |
| LCNN | $77.16 \pm .07$ | $61.17 \pm .53$ | $39.72 \pm .17$ | $25.60 \pm .32$ | $17.66 \pm .18$ |
| GradReg | $77.20 \pm .26$ | $71.90 \pm .11$ | $61.06 \pm .03$ | $49.19 \pm .12$ | $38.09 \pm .47$ |
| LCNNs + GradReg | $77.29 \pm .26$ | $\mathbf{72.68} \pm .52$ | $\mathbf{63.36} \pm .39$ | $\mathbf{52.96} \pm .76$ | $\mathbf{42.70} \pm .77$ |
| CURE [6] | $76.48 \pm .07$ | $71.39 \pm .12$ | $61.28 \pm .32$ | $49.60 \pm .09$ | $39.04 \pm .16$ |
| Softplus + Wt. Decay [7] | $77.44 \pm .28$ | $60.86 \pm .36$ | $38.04 \pm .43$ | $23.85 \pm .33$ | $16.20 \pm .01$ |
| Adversarial Training [32] | $76.96 \pm .26$ | $\mathbf{72.76} \pm .15$ | $\mathbf{64.70} \pm .20$ | $\mathbf{54.80} \pm .25$ | $\mathbf{44.98} \pm .57$ |

## 5.4    Train-Test Discrepancy in Model Geometry

During our experiments, we observed a consistent phenomenon where the gradient norms and Hessian norms for test data points were much larger than those for the train data points, which hints at a form of overfitting with regards to these quantities. We term this phenomenon as the *train-test discrepancy* in model geometry. Interestingly, we did not observe any such discrepancy using our proposed curvature measure, indicating that our proposed measure may be a more reliable measure of model geometry. We report our results in Table 3, where we measure the relative discrepancy, finding that the discrepancy for our proposed measure of curvature is multiple orders of magnitude smaller than the corresponding quantity for gradient and Hessian norms. We leave further investi-

gation of this phenomenon – regarding why curvature is stable across train and test – as a topic for future work.

Table 3: Train-test descrepancy in model geometry, where the relative descrepancy $\Delta_{tt}g(\mathcal{X}) = \left| \frac{g(\mathcal{X}_{\text{test}}) - g(\mathcal{X}_{\text{train}})}{g(\mathcal{X}_{\text{test}})} \right|$ is shown for three different geometric measures. We observe that (1) there exists a large train-test descrepancy, with the test gradient / hessian norms being $> 10\times$ the corresponding values for the train set. (2) the descrepancy is 2-3 orders of magnitude smaller for our proposed curvature measure, indicating that it may be a stable model property.

| **Model** | $\Delta_{tt}\mathbb{E}_{\mathbf{x}\in\mathcal{X}}\|\nabla f(\mathbf{x})\|_2$ | $\Delta_{tt}\mathbb{E}_{\mathbf{x}\in\mathcal{X}}\|\nabla^2 f(\mathbf{x})\|_2$ | $\Delta_{tt}\mathbb{E}_{\mathbf{x}\in\mathcal{X}}\mathcal{C}_f(\mathbf{x})$ |
|---|---|---|---|
| Standard | 11.75 | 12.28 | **0.025** |
| GradReg | 11.33 | 11.22 | **0.017** |
| LCNN | 19.99 | 11.33 | **0.129** |
| LCNNs + GradReg | 21.82 | 10.43 | **0.146** |

**Summary of Experimental Results**  Overall, our experiments show that:

(1) LCNNs have lower curvature than standard models as advertised, and combining them with gradient norm regularization further decreases curvature (see Table 1). The latter phenomenon is unexpected, as our curvature measure ignores gradient scaling.

(2) LCNNs combined with gradient norm regularization achieve an order of magnitude improved gradient robustness over standard models (see Figure 2).

(3) LCNNs combined with gradient norm regularization outperform adversarial training in terms of achieving a better predictive accuracy at a lower curvature (see Table 1), and are competitive in terms of adversarial robustness (see Table 2), while being $\sim 1.4\times$ faster.

(4) We observe that there exists a train-test discrepancy for standard geometric quantities like the gradient and Hessian norm, and this discrepancy disappears for our proposed curvature measure (see Table 3).

We also present ablation experiments, additional adversarial attacks, and evaluations on more datasets and architectures in the Appendix.

## 6    Discussion

In this paper, we presented a modular approach to remove excess curvature in neural network models. Importantly, we found that combining vanilla LCNNs with gradient norm regularization resulted in models with the smallest curvature, the most stable gradients as well as those that are the most adversarially robust. Notably, this procedure achieves adversarial robustness **without** explicitly generating adversarial examples during training.

The current limitations of our approach are that we only consider convolutional and fully connected layers, and not self-attention or recurrent layers. We also do not investigate the learning-theoretic benefits (or harms) of low-curvature models, or study their generalization for small number of training samples, or their robustness to label noise (which we already observe in Fig. 1b). Investigating these are important topics for future work.

## Acknowledgments and Disclosure of Funding

The authors would like to thank the anonymous reviewers for their helpful feedback and all the funding agencies listed below for supporting this work. SS and HL are supported in part by the NSF awards #IIS-2008461 and #IIS-2040989, and research awards from Google, JP Morgan, Amazon, Harvard Data Science Initiative, and D$^3$ Institute at Harvard. KM and SS (partly) are supported by the Swiss National Science Foundation under grant number FNS-188758 "CORTI". HL would like to thank Sujatha and Mohan Lakkaraju for their continued support and encouragement.

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
