# Appendix: Efficient Training of Low-Curvature Neural Networks

## A Proof of Theorem 1

In this section, we provide the proof of the main theorem in the paper, which decomposes overall curvature into curvatures and slopes of constituent layers. We state Theorem 1 below for reference.

**Theorem 1.** *Given a function $f = f_L \circ f_{L-1} \circ \ldots \circ f_1$ with $f_i : \mathbb{R}^{n_{i-1}} \to \mathbb{R}^{n_i}$, the curvature $\mathcal{C}_f$ can be bounded as follows*

$$\mathcal{C}_f(\mathbf{x}) \leq \sum_{i=1}^{L} n_i \times \mathcal{C}_{f_i}(\mathbf{x}) \prod_{j=1}^{i} \|\nabla_{f_{j-1}} f_j(\mathbf{x})\|_2 \leq \sum_{i=1}^{L} n_i \times \max_{\mathbf{x}'} \mathcal{C}_{f_i}(\mathbf{x}') \prod_{j=1}^{i} \max_{\mathbf{x}'} \|\nabla_{f_{j-1}} f_j(\mathbf{x}')\|_2. \tag{1}$$

This statement is slightly different than the one given in the paper, differing by a term of the width of each nonlinear layer. Since we do not use or care about the units of curvature, only its minimization, and have elected to equally-weight each term of the sum, this is an inconsequential discrepancy.

A similar bound is constructed recursively in [1]. [2] gives a similar formula, albeit for the Frobenius norm. The Frobenius norm is both simpler, because the sum of squared entries is independent of the layout of the data, and also weaker, since it cannot deliver a bound which holds uniformly in the data. To our knowledge Equation 1 is the first explicit, easily-interpreted formula of its type.

We start with some preliminaries, with the actual proof being in subsection A.4.

### A.1 Derivatives of compositional functions

For a function $f : \mathbb{R}^d \to \mathbb{R}^r$, let $\nabla f : \mathbb{R}^d \to \mathbb{R}^{d \times r}$ denote its gradient, and $\nabla^2 f : \mathbb{R}^d \to \mathbb{R}^{d \times r \times d}$ denote its Hessian. We drop the argument to functions when possible, and all norms will be spectral norms.

Given $L$ functions $f_i : \mathbb{R}^{n_{i-1}} \to \mathbb{R}^{n_i}, i = 1 \ldots, L$ let $f_{k,k+j} = f_{k+j} \circ f_{k+j-1} \circ \ldots \circ f_k : \mathbb{R}^{n_{k-1}} \to \mathbb{R}^{n_{k+j}}$ for $1 \leq k \leq k + j \leq L$. This function composition will be our model of a deep neural network where $f_\ell$ represents the action of the $\ell$th layer.

If each $f_i$ is continuously differentiable, we have this formula for the gradient of $f_{k,k+j}$

$$\nabla f_{k,k+j} = \prod_{i=1}^{j} \nabla f_{k+j-i+1} \in \mathbb{R}^{n_{k+j} \times n_k}. \tag{2}$$

where we adopt the convention that $f_{j,j}(x) = x$ in order to make the boundary conditions correct. The product begins at the end, with $\nabla f_{k+j}$ and progresses forward through the indices – a straightforward consequence of the chain rule of differentiation. Supposing moreover that each $f_i$ is twice-differentiable, we have this formula for the second derivative of $f_{1,k}$:

$$\nabla^2 f_{1,k} = \sum_{i=1}^{k} \nabla^2 f_{1,i} - \nabla^2 f_{1,i-1} \tag{3}$$

$$\text{where } (\nabla^2 f_{1,i} - \nabla^2 f_{1,i-1}) = (\nabla^2 f_i)\left(\nabla f_{1,i-1}, \nabla f_{i+1,k}^{\top}, \nabla f_{1,i-1}\right) \in \mathbb{R}^{n_0 \times n_k \times n_0}$$

where we have used the *covariant multilinear matrix multiplication* notation: $\nabla^2 f_i$ is an order-three tensor $\in \mathbb{R}^{n_i \times n_i \times n_i}$, with the first and third modes multiplied by $\nabla f_{1,i-1} \in \mathbb{R}^{n_i \times n_0}$ and the second mode multiplied by $\nabla f_{i+1,k}^{\top} \in \mathbb{R}^{n_i \times n_k}$.

## A.2 Tensor calculus

In this section, we present a simplifed version of the notation from [3]. A $k$-linear map is a function of $k$ (possibly multidimensional) variables such that if any $k-1$ variables are held constant, the map is linear in the remaining variable. A $k$-linear function is can be represented by an order-$k$ tensor $\mathcal{A}$ given elementwise by $\mathcal{A} = [\![a_{j_1 \dots j_k}]\!] \in \mathbb{R}^{d_1 \times \dots \times d_k}$.

The covariant multilinear matrix multiplication of a tensor with matrices (order 2 tensors) $M_1 = (m_{i_1 j_1}^{(1)}) \in \mathbb{R}^{d_1 \times s_1}, \dots, M_k = (m_{i_k j_k}^{(k)}) \in \mathbb{R}^{d_k \times s_k}$ is

$$\mathcal{A}(M_1, \dots, M_k) = \left[\!\!\left[\sum_{i_1=1}^{d_1} \dots \sum_{i_k}^{d_k} a_{i_1 \dots i_k} m_{i_1 j_1}^{(1)} \dots m_{i_k j_k}^{(k)}\right]\!\!\right] \in \mathbb{R}^{s_1 \times \dots \times s_k}.$$

This operation can be implemented via iterated einsums as:

```
def covariant_multilinear_matmul(a: torch.Tensor,
                                 mlist: List[torch.Tensor]) -> torch.Tensor:
    order = a.ndim
    base_indices = string.ascii_letters
    indices = base_indices[:order]
    next_index = base_indices[order]

    val = a
    for idx in range(order):
        resp_str = indices[:idx] + next_index + indices[idx+1:]
        einsum_str = indices + f",{indices[idx]}{next_index}->{resp_str}"
        val = torch.einsum(einsum_str, val, mlist[idx])
    return val
```

For example, covariant multilinear matrix multiplication of an order two tensor is pre- and post-multiplication by its arguments: $M_1^{\top} \mathcal{A} M_2 = \mathcal{A}(M_1, M_2)$. The generalization of the matrix spectral norm is $||\mathcal{A}||_2 = \sup\{\mathcal{A}(x_1, \dots, x_k) : ||x_i|| = 1, x_i \in \mathbb{R}^{d_i}, i = 1, 2, \dots, k\}$. The computation of order-$k$ operator norms is hard in theory, and also in practice (cf. [4]). In order to address this difficulty, we introduce an instance of the *unfold* operator.

For $\mathcal{A} \in \mathbb{R}^{d_1 \times d_2 \times d_3}$, $\text{unfold}_{\{\{1,2\},\{3\}\}}(\mathcal{A}) \in \mathbb{R}^{d_1 d_2 \times d_3}$ is the matrix with the $j$th column being the flattened $j$th (in the final index) $d_2 \times d_3$ matrix.[1] Unfolding is useful because it it allows us to bound an order-3 operator norm in terms of order-2 operator norms – Wang et al. [3] shows that $||\mathcal{A}|| \le ||\text{unfold}_{\{\{1,2\},\{3\}\}}(\mathcal{A})||$. The upper bound – the operator norm of a *matrix* – can computed with standard largest singular-value routines. A similar unfolding-based bound was used in the deep-learning context by [5] to give improved estimates on the spectral norm of convolution operators.

To facilitate the analysis of unfolded tensors, we coin operations that *put to* the diagonal of tensors:

- $\text{pdiag2} : \mathbb{R}^d \mapsto \mathbb{R}^{d \times d}$ defined by $\text{pdiag2}(x)_{ij} = \begin{cases} x_j & \text{if } i = j \\ 0 & \text{otherwise.} \end{cases}$

---

[1] In PyTorch notation, $\text{unfold}_{\{\{1,2\},\{3\}\}}(\text{a}) = \text{torch.flatten(a, end\_dim=1)}$

- pdiag3 : $\mathbb{R}^d \mapsto \mathbb{R}^{d \times d \times d}$ defined by $\mathrm{pdiag3}(x)_{ijk} = \begin{cases} x_j & \text{if } i = j = k \\ 0 & \text{otherwise.} \end{cases}$

Further, let $1_n \in \mathbb{R}^n$ be a vector of ones, $I_n = \mathrm{pdiag2}(1_n) \in \mathbb{R}^{n \times n}$ be the $n$-dimensional identity matrix, and $\mathcal{I}_n = \mathrm{pdiag3}(1_n) \in \mathbb{R}^{n \times n \times n}$. For two vectors $a \in \mathbb{R}^n, b \in \mathbb{R}^n$, let $ab$ denote the elementwise product. $\otimes$ denotes the well-understood Kronecker product, so that, for example, $1_n^\top \otimes I_m$ is an $m \times nm$ matrix consisting of $n$ copies of the $m \times m$ identity matrix stacked side by side. Where it is redundant, we drop the subscripts indicating dimension.

We use the following facts about tensors, their unfoldings, and their operator norms, in what follows.

1. $\mathcal{A} = \mathrm{pdiag3}(ab) \implies \mathcal{A}(M_1, M_2, M_3) = \mathcal{I}(\mathrm{pdiag2}(a)M_1, M_2, \mathrm{pdiag2}(b)M_3)$

2. $\mathrm{unfold}_{\{\{1,2\},\{3\}\}}(\mathcal{I}(M_1, M_2, M_3)) = (M_1 \otimes I_{s_2})^\top \mathrm{unfold}_{\{\{1,2\},\{3\}\}}(\mathcal{I}(I_{d_1}, M_2, M_3))$

3. $||(1_{d_1}^\top \otimes I_{s_2})\mathrm{unfold}_{\{\{1,2\},\{3\}\}}(\mathcal{I}(I_{d_1}, M_2, M_3))|| \leq ||M_2^\top M_3|| s_2$

Taken together Fact #2 and #3 imply that

$$||\mathrm{unfold}_{\{\{1,2\},\{3\}\}}(\mathcal{I}(M_1, M_2, M_3))|| \leq ||M_1|| \times ||M_2^\top M_3|| \times s_2 \tag{4}$$

which is our essential bound for the norm of an order-3 tensor in terms of order-2 tensors.

## A.3 Hessian increment bound

Let $\sigma(x) = \exp(x)/(1 + \exp(x)))$ denote the (elementwise) logistic function $\mathbb{R}^d \mapsto \mathbb{R}^d$. The derivatives of $s(x; \beta)$ can be written as

$$\nabla s(x; \beta) = \mathrm{pdiag2}(\sigma(\beta x)) \in \mathbb{R}^{d \times d} \tag{5}$$
$$\nabla^2 s(x; \beta) = \mathrm{pdiag3}(\beta \sigma(\beta x)(1 - \sigma(\beta x))) \in \mathbb{R}^{d \times d \times d}. \tag{6}$$

Let the $i$th softplus layer have coefficient $\beta_i$, then the increment from Equation 3, can be bounded as follows:

$$||(\nabla^2 f_i)\left(\nabla f_{1,i-1}, \nabla f_{i+1,k}^\top, \nabla f_{1,i-1}\right)|| \tag{7}$$
$$=||\beta_i \mathcal{I}_{n_i}\left(\mathrm{pdiag2}(1 - \sigma(\beta_i x))\nabla f_{1,i-1}, \nabla f_{i+1,k}^\top, \mathrm{pdiag2}(\sigma(\beta_i x))\nabla f_{1,i-1}\right)|| \tag{8}$$
$$\leq||\beta_i \mathrm{unfold}_{\{\{1,2\},\{3\}\}}(\mathcal{I}_{n_i}(\mathrm{pdiag2}(1 - \sigma(\beta_i x))\nabla f_{1,i-1}, \nabla f_{i+1,k}^\top, \mathrm{pdiag2}(\sigma(\beta_i x))\nabla f_{1,i-1})|| \tag{9}$$
$$\leq||\beta_i(\mathrm{pdiag2}(1 - \sigma(\beta_i x))\nabla f_{1,i-1}|| \times ||\nabla f_{i+1,k}^\top \mathrm{pdiag2}(\sigma(\beta_i x))\nabla f_{1,i-1}|| \times n_i \tag{10}$$
$$\leq n_i \times ||\beta_i(\mathrm{pdiag2}(1 - \sigma(\beta_i x)))|| \times ||\nabla f_{1,i-1}|| \times ||\nabla f_{i+1,k}^\top \nabla f_i \nabla f_{1,i-1}|| \tag{11}$$
$$= n_i \times \mathcal{C}_{f_i} \times ||\nabla f_{1,i-1}|| \times ||\nabla f||. \tag{12}$$

Equation 8 follows by Fact #1 above, along with Equation 6. Equation 9 is the standard unfolding bound by [3]. Equation 10 is our main bound on order-3 tensors in terms of order-2 matrices, Equation 4. Equation 11 follows from the Cauchy-Schwartz inequality. The replacement in the last term of the product is Equation 5. Equation 12 rewrites Equation 11 using Equation 2: $f_{i+1,k}^\top \nabla f_i \nabla f_{1,i-1} = \nabla f_{1,k}$ and the definition of the curvature of $f_i$.

## A.4 Putting it together

Ignoring the $\epsilon$ term,

$$\mathcal{C}_f(\mathbf{x}) = \frac{||\nabla^2 f(\mathbf{x})||}{||\nabla f(\mathbf{x})||} \tag{13}$$

$$= \frac{1}{||\nabla f(\mathbf{x})||} \left|\left| \sum_{i=1}^{L} \nabla^2 f_{1,i}(\mathbf{x}) - \nabla^2 f_{1,i-1}(\mathbf{x}) \right|\right| \tag{14}$$

$$\leq \frac{1}{||\nabla f(\mathbf{x})||} \sum_{i=1}^{L} ||\nabla^2 f_{1,i}(\mathbf{x}) - \nabla^2 f_{1,i-1}(\mathbf{x})|| \tag{15}$$

$$\leq \sum_{i=1}^{L} n_i \times \mathcal{C}_{f_i}(\mathbf{x}) \times ||\nabla f_{1,i-1}(\mathbf{x})|| \tag{16}$$

$$\leq \sum_{i=1}^{L} n_i \times \mathcal{C}_{f_i}(\mathbf{x}) \times \prod_{i=1}^{j} ||\nabla f_i(\mathbf{x})|| \tag{17}$$

$$\leq \sum_{i=1}^{L} n_i \times \max_{\mathbf{x}} \mathcal{C}_{f_i}(\mathbf{x}) \times \prod_{i=1}^{j} \max_{\mathbf{x}} ||\nabla f_i(\mathbf{x})|| \tag{18}$$

Equation 14 substitutes Equation 3. Equation 15 is the triangle inequality. Equation 16 is Equation 12, along with cancelling the term of $||\nabla f||$ top and bottom. Equation 17 are Equation 18 are obvious and a standard simplification in the literature on controlling the Lipschitz constant of neural networks. Because exactly computing the smallest Lipschitz constant of a general neural network is NP-complete, a widely-used baseline measure of Lipschitz-smoothness is rather the product of the Lipschitz constants of smaller components of the network, such as single layers ([6]). □

# B Loss curvature vs logit curvature

We have thusfar discussed how to assure a bound on curvature independent of the input $x$. However, a truly data-independent bound must also not depend on the class label $y$. In this short section we discuss the main consideration that accompanies this: the differences between loss and (pre-softmax) logit curvature. Let the function including the loss for a class $c$ be denoted by $f_{\text{loss}}^c$, and the logit corresponding to a class $c \in [1, C]$ be $f_{\text{logit}}^c$, and let $f_{\text{logit}}^{1,C}$ denote the vector valued function corresponding to all the logits. Then we have $f_{\text{loss}}^c = \text{lsm}^c \circ f_{\text{logit}}^{1,C}$, where $\text{lsm}^c(\mathbf{x}) = -\log \frac{\exp(\mathbf{x}_c)}{\sum_{i=1}^C \exp(\mathbf{x}_i)} = -\mathbf{x}_c + \log \sum_{i=1}^C \exp(\mathbf{x}_i)$ is the negative log softmax function.

The derivatives of this function are:

$$\frac{d\text{lsm}^c(\mathbf{x})}{d\mathbf{x}_c} = \frac{\exp(\mathbf{x}_c)}{\sum_{i=1}^C \exp(\mathbf{x}_i)} - 1$$

$$\frac{d^2\text{lsm}^c(\mathbf{x})}{d\mathbf{x}_c^2} = \frac{\exp(\mathbf{x}_c)}{\sum_{i=1}^C \exp(\mathbf{x}_i)} \left(1 - \frac{\exp(\mathbf{x}_c)}{\sum_{i=1}^C \exp(\mathbf{x}_i)}\right)$$

whose norm is upper bounded by 1 and 0.25 respectively. This implies that bounding the logit curvature also ensures that the loss curvature is bounded, as the gradients and Hessians of the negative log softmax layer do not explode. In other words, $\mathcal{C}_{f_{\text{loss}}} < \mathcal{C}_{f_{\text{logit}}}$.

However penalizing an upper bound may not be the most efficient way to penalize the loss Hessian. Consider the following decomposition of the loss Hessian, which can be written as $\nabla^2 f(\mathbf{x}) \sim \nabla f_{\text{logit}}(\mathbf{x}) \nabla_{f_l}^2 \text{LSE}(\mathbf{x}) \nabla f_{\text{logit}}(\mathbf{x})^\top + \nabla_{f_l} \text{LSE}(\mathbf{x}) \nabla^2 f_{\text{logit}}(\mathbf{x})$, where $\text{LSE}(\mathbf{x})$ is the Log-SumExp function. Thus a more efficient way to penalize the loss Hessians is to both use our LCNN penalties, as well as penalize the gradient norm, which we also find to be true in our experiments.

## C γ-Lipschitz Batchnorm

We present here a Pytorch-style pseudo-code for the γ-Lipschitz Batchnorm for clarity.

```
bn = torch.nn.BatchNorm2d(in_channels, affine=False)
log_lipschitz = torch.nn.Parameter(torch.tensor(init_lipschitz).log())
...
# perform spectral normalization for BN in closed form
bn_spectral_norm = torch.max(1 / (bn.running_var + 1e-5).sqrt())
one_lipschitz_bn = bn(x) / bn_spectral_norm

# multiply normalized BN with a learnable scale
scale = torch.min((bn.running_var + 1e-5) ** .5)
one_lipschitz_part = bn(x) * scale
x = one_lipschitz_part * torch.minimum(1 / scale, log_lipschitz.exp())
```

Note that we parameterize $\gamma$ in terms of its log value, to ensure it remains positive during training. We employ the same method for fitting the centered softplus parameter, $\beta$.

## D Proofs of LCNN Properties

Here we present proofs for the properties linking LCNNs to gradient smoothness and adversarial robustness. To this end, we first prove a lemma that is used in both these results, which states that the gradient norm can rise exponentially in the worst case. In the text below, $B_\delta(\mathbf{x})$ denotes the $\delta$-ball around $\mathbf{x}$.

**Lemma 1.** *The ratio of gradient norms at nearby points rises exponentially with distance between those points. For a function $f$ satisfying $\max_{\mathbf{x}' \in B_\delta(\mathbf{x})} \mathcal{C}_f(\mathbf{x}') = \delta_\mathcal{C}$, points $\mathbf{x}$ and $\mathbf{x} + \epsilon$ that are $\|\epsilon\|_2 = r$ away, we have*

$$\frac{\|\nabla f(\mathbf{x} + \epsilon)\|_2}{\|\nabla f(\mathbf{x})\|_2} \leq \exp(r\delta_\mathcal{C})$$

*Proof.* Let $g(\mathbf{x}) = \log \|\nabla f(\mathbf{x})\|_2^2$. Applying a first order Taylor series with Lagrange remainder / mean-value theorem, we have

$$
\begin{aligned}
g(\mathbf{x} + \epsilon) - g(\mathbf{x}) &= \nabla g(\mathbf{x} + \xi)^\top \epsilon && \text{(for some } \xi \in B_\epsilon(\mathbf{x})\text{)} \\
\log \frac{\|\nabla f(\mathbf{x} + \epsilon)\|_2^2}{\|\nabla f(\mathbf{x})\|_2^2} &= 2 \frac{\nabla f(\mathbf{x} + \xi) \nabla^2 f(\mathbf{x} + \xi)^\top \epsilon}{\|\nabla f(\mathbf{x} + \xi)\|_2^2} \\
&\leq 2 \frac{\|\nabla f(\mathbf{x} + \xi)\|_2 \|\nabla^2 f(\mathbf{x} + \xi)\|_2 \|\epsilon\|_2}{\|\nabla f(\mathbf{x} + \xi)\|_2^2} && \text{(Cauchy-Schwartz inequality)} \\
&\leq 2r\delta_\mathcal{C}
\end{aligned}
$$

Note that to apply the Cauchy-Schwartz inequality, we first upper bound the term on the right by its 2-norm, which for a scalar is simply its absolute value. Taking exponent on both sides of the final expression and taking square root, we have the intended result.

□

### D.1 LCNNs have Robust Gradients

**Proposition 1.** *Let $\max_{\mathbf{x}' \in B_\epsilon(\mathbf{x})} \mathcal{C}_f(\mathbf{x}') \leq \delta_\mathcal{C}$, then the relative distance between gradients at $\mathbf{x}$ and $\mathbf{x} + \epsilon$ is*

$$\frac{\|\nabla f(\mathbf{x} + \epsilon) - \nabla f(\mathbf{x})\|_2^2}{\|\nabla f(\mathbf{x})\|_2^2} \leq r\delta_\mathcal{C} \exp(r\delta_\mathcal{C}) \sim r\mathcal{C}_f(\mathbf{x}) \quad \textit{(Quadratic Approximation)}$$

*Proof.* We begin the proof by invoking the Taylor series approximation of $\nabla f(\mathbf{x}+\epsilon)$ at $\mathbf{x}$, and using the explicit Lagrange form of the Taylor error. This is equivalent to using a form of the multivariate mean-value theorem. Let $\xi \in B_\epsilon(\mathbf{x})$, then there exists some $\xi$ such that the following holds

$$\nabla f(\mathbf{x} + \epsilon) - \nabla f(\mathbf{x}) = \nabla^2 f(\mathbf{x} + \xi)^\top \epsilon \qquad \text{(Taylor Series)}$$

$$\|\nabla f(\mathbf{x} + \epsilon) - \nabla f(\mathbf{x})\|_2 \le \|\nabla^2 f(\mathbf{x} + \xi)\|_2 \|\epsilon\|_2 \qquad \text{(Cauchy- Schwartz inequality)}$$

$$\frac{\|\nabla f(\mathbf{x} + \epsilon) - \nabla f(\mathbf{x})\|_2}{\|\nabla f(\mathbf{x})\|_2} \le \frac{\mathcal{C}_f(\mathbf{x} + \xi_0)\|\nabla f(\mathbf{x} + \xi)\|_2}{\|\nabla f(\mathbf{x})\|_2} r \qquad (\text{Divide by gradnorm})$$

Plugging in the value of $\|\nabla f(\mathbf{x} + \xi_0)\|_2 / \|\nabla f(\mathbf{x})\|_2$ from Lemma 1, and further upper bounding $\mathcal{C}_f(\mathbf{x} + \xi) \le \delta_{\mathcal{C}}$ we have the intended result.

To derive the simplified quadratic approximation, replace the first step in the Taylor series with $\nabla f(\mathbf{x} + \epsilon) - \nabla f(\mathbf{x}) = \nabla^2 f(\mathbf{x})^\top \epsilon$, i.e., use $\xi = 0$. $\qquad\square$

### D.2 Curvature is Necessary for Robustness

**Proposition 2.** *Let $\max_{\mathbf{x}' \in B_\epsilon(\mathbf{x})} \mathcal{C}_f(\mathbf{x}') \le \delta_{\mathcal{C}}$, then for two points $\mathbf{x}$ and $\mathbf{x} + \epsilon$,*

$$\|f(\mathbf{x} + \epsilon) - f(\mathbf{x})\|_2 \le r\|\nabla f\|_2 \left(1 + \frac{1}{2}r\delta_{\mathcal{C}} \exp(r\delta_C)\right) \sim r\|\nabla f\|_2 \left(1 + \frac{1}{2}r\mathcal{C}_f(\mathbf{x})\right) \quad (\text{Quadratic Approximation})$$

*Proof.* Let $\xi \in B_\epsilon(\mathbf{x})$, then there exists some $\xi$ such that the following holds

$$f(\mathbf{x} + \epsilon) - f(\mathbf{x}) = \nabla f(\mathbf{x})^\top \epsilon + \frac{1}{2}\epsilon^\top \nabla^2 f(\mathbf{x} + \xi)\epsilon \qquad \text{(Taylor Series)}$$

$$\|f(\mathbf{x} + \epsilon) - f(\mathbf{x})\|_2 \le \|\nabla f(\mathbf{x})^\top \epsilon\|_2 + \frac{1}{2}\|\epsilon^\top \nabla^2 f(\mathbf{x} + \xi)\epsilon\|_2 \qquad \text{(triangle inequality)}$$

$$\le \|\nabla f(\mathbf{x})\|_2 \|\epsilon\|_2 + \frac{1}{2}\lambda_{\max}(\mathbf{x} + \xi)\|\epsilon\|_2^2 \qquad \text{(Cauchy-Schwartz inequality)}$$

$$\le \|\nabla f(\mathbf{x})\|_2 \|\epsilon\|_2 + \frac{1}{2}\mathcal{C}_f(\mathbf{x} + \xi)\|\epsilon\|_2^2 \|\nabla f(\mathbf{x} + \xi)\|_2 \qquad (\text{Defn of } \mathcal{C}_f(\mathbf{x} + \xi))$$

Factoring out $\|\nabla f(\mathbf{x})\|_2 \|\epsilon\|_2$ in the RHS, and using Lemma 1, and further upper bounding $\mathcal{C}_f(\mathbf{x} + \xi) \le \delta_{\mathcal{C}}$ we have the intended result.

To derive the simplified quadratic approximation, replace the first step in the Taylor series with $f(\mathbf{x} + \epsilon) - f(\mathbf{x}) = \nabla f(\mathbf{x})^\top \epsilon + \frac{1}{2}\epsilon^\top \nabla^2 f(\mathbf{x})\epsilon$, i.e., use $\xi = 0$.

$\qquad\square$

## E Experimental Settings

In this section we elaborate on the hyper-parameter settings used for our tuning our models. For the standard ResNet-18, we use standard hyper-parameter settings as indicated in the main paper, and we do not modify this for the other variants. For LCNNs, we chose regularizing constants as $\lambda_\beta = 10^{-4}$ and $\lambda_\gamma = 10^{-5}$. For GradReg, we use $\lambda_{\text{grad}} = 10^{-3}$, and for LCNNs + GradReg, we chose $\lambda_\beta = 10^{-4}$, $\lambda_\gamma = 10^{-5}$, $\lambda_{\text{grad}} = 10^{-3}$. We performed a coarse grid search and chose the largest regularizing constants that did not affect predictive performance.

## F Additional Experiments

### F.1 Ablation Experiments

In this section, we present ablation studies where we train models with each of the proposed modifications separately, i.e., we train a model with only spectral norm for the convolution layers, $\gamma$-Lipschitz Batchnorm or centered softplus. Our results in Table 1 show that for the resnet18 architecture considered,

Table 1: Ablation experiments to study effect of individual modifications to LCNN architectures. We find that while either using only centered softplus or $\gamma$-BN suffices in practice to minimize curvature, while spectral norm on the convolutional layers (which is the most expensive modification) may not be necessary.

| Model | $\mathbb{E}_{\mathbf{x}}\mathcal{C}_f(\mathbf{x})$ | $\mathbb{E}_{\mathbf{x}}\|\nabla^2 f(\mathbf{x})\|_2$ | $\mathbb{E}_{\mathbf{x}}\|\nabla f(\mathbf{x})\|_2$ | Accuracy (%) |
|---|---|---|---|---|
| ConvSpectralNorm only | 358.86 | 8380.55 | 23.92 | 77.55 |
| $\gamma$-BN only | 65.78 | 1086.95 | 20.86 | 77.33 |
| c-Softplus only | 57.49 | 734.05 | 16.99 | 77.31 |
| Standard | 270.89 | 6061.96 | 19.66 | 77.42 |
| LCNN | 69.40 | 1143.62 | 22.04 | 77.30 |

(1) performing spectral normalization had no effect of curvature as presumably the batchnorm layers are able to compensate for the lost flexibility, and

(2) either penalizing the batchnorm alone or the softplus alone performs almost as well as LCNN, or sometimes even better in terms of curvature reduction. Note that the most expensive computational step is the spectral norm for the convolutional layers, indicating that avoiding this in practice may yield speedups in practice.

While in practice for Resnet-18 only $\gamma$-Lipschitz or centered softplus is sufficient for curvature reduction, in theory we must regularize all components to avoid scaling restrictions in one layer being compensated by other layers, as dictated by the upper bound. In particular, this means that the strategy of penalizing only a subset of layers may not generalize to other architectures.

## F.2 Robustness Evaluation on RobustBench / Autoattack

The attack presented in **??** was relatively "weak" – a network could be truly susceptible to adversarial attack, but by only testing against a weak adversary, we could fail to notice. Short of a comprehensive verification (e.g. [7]), which is known to be computationally intractable at scale, there is no fully satisfactory way to guarantee robustness. However, one common method to develop confidence in a model is to demonstrate robustness in the face of a standard set of nontrivially capable attackers. This is what we do in Table 2, where we use the Robustbench software library [8] to evaluate both a white-box (having access to the internal details of the model), and a black-box (using only function evaluations) attacks.

Table 2: Adversarial accuracy to standard attacks accessed via Robustbench [8]. "APGD-t" refers to the white box targeted auto PGD attack from [9], "Square" refers to the black-box square attack from [10].

| Model | APGD-t Acc. (%) | Square Acc. (%) | Clean Acc. (%) |
|---|---|---|---|
| Standard | 22.122 | 52.874 | 76.721 |
| LCNN | 23.709 | 52.179 | 76.602 |
| GradReg | 50.294 | 64.678 | 76.394 |
| LCNN+GradReg | 52.477 | 64.678 | 76.622 |
| CURE | 50.096 | 63.488 | 75.928 |
| Softplus+wt decay | 23.907 | 53.766 | 76.622 |
| Adv Training | 55.155 | 66.861 | 75.521 |
| CURE + GradReg | 60.810 | 67.357 | 74.192 |
| LCNN + GradReg + Adv Training | 59.222 | 66.861 | 75.382 |

These experimental results show overall that:

(1) LCNN + GradReg is still on par with adversarial training even against stronger attacks such as APGD-t and Square, as they are with PGD.

(2) Combining LCNN + GradReg with adversarial training (in the last row) further improves robustness at the cost of predictive accuracy.

(3) Combining CURE with GradReg (in the penultimate row) improves robustness at the cost of further deteriorating predictive accuracy.

### F.3 Additional Evaluations on more Architectures / Dataset Combinations

We present results on the following dataset - architecture pairs:

(1) In Table 3, we present results on SVHN dataset and VGG-11 model

(2) In Table 4, we present results on SVHN dataset and ResNet-18 model

(3) In Table 5, we present results on CIFAR-100 dataset and VGG-11 model

In all cases, we find that our results are on par with our experiment done on the CIFAR-100 and ResNet-18 setup, which confirms that generality of our approach.

Table 3: Model geometry of VGG-11 models trained on the SVHN test dataset.

| Model | $\mathbb{E}_{\mathbf{x}}\|\nabla f(\mathbf{x})\|_2$ | $\mathbb{E}_{\mathbf{x}}\|\nabla^2 f(\mathbf{x})\|_2$ | $\mathbb{E}_{\mathbf{x}}\mathcal{C}_f(\mathbf{x})$ | Accuracy (%) |
|---|---|---|---|---|
| Standard | 2.87 | 158.29 | 54.24 | 96.01 |
| LCNNs | 4.04 | 83.34 | 30.05 | 95.61 |
| GradReg [11] | 1.85 | 57.52 | 33.34 | 96.03 |
| LCNNs + GradReg | 2.02 | 25.54 | 17.06 | 96.23 |
| Adversarial Training [12] | 1.25 | 27.64 | 24.23 | 96.37 |

Table 4: Model geometry of Resnet-18 models trained on the SVHN test dataset.

| Model | $\mathbb{E}_{\mathbf{x}}\|\nabla f(\mathbf{x})\|_2$ | $\mathbb{E}_{\mathbf{x}}\|\nabla^2 f(\mathbf{x})\|_2$ | $\mathbb{E}_{\mathbf{x}}\mathcal{C}_f(\mathbf{x})$ | Accuracy (%) |
|---|---|---|---|---|
| Standard | 2.64 | 204.38 | 78.22 | 96.41 |
| LCNNs | 2.91 | 77.78 | 25.36 | 96.35 |
| GradReg [11] | 1.63 | 68.22 | 39.55 | 96.57 |
| LCNNs + GradReg | 1.69 | 31.69 | 15.28 | 96.53 |
| Adversarial Training [12] | 1.05 | 22.96 | 24.48 | 96.64 |

Table 5: Model geometry of VGG-11 models trained on the CIFAR-100 test dataset.

| Model | $\mathbb{E}_{\mathbf{x}}\|\nabla f(\mathbf{x})\|_2$ | $\mathbb{E}_{\mathbf{x}}\|\nabla^2 f(\mathbf{x})\|_2$ | $\mathbb{E}_{\mathbf{x}}\mathcal{C}_f(\mathbf{x})$ | Accuracy (%) |
|---|---|---|---|---|
| Standard | 17.07 | 1482.16 | 85.81 | 73.33 |
| LCNNs | 15.88 | 282.06 | 41.14 | 73.76 |
| GradReg [11] | 10.64 | 534.71 | 48.26 | 72.65 |
| LCNNs + GradReg | 9.81 | 105.07 | 24.48 | 73.01 |
| Adversarial Training [12] | 6.20 | 166.73 | 27.37 | 71.13 |