# OpenReview forum: "Efficient Training of Low-Curvature Neural Networks"
_NeurIPS.cc/2022/Conference — NeurIPS 2022 Accept_

### Official Review · Reviewer_1uxf · 2022-07-03

**Rating:** 4
**Confidence:** 4
**Soundness:** 2 fair
**Presentation:** 3 good
**Contribution:** 2 fair

**Summary:**

This paper proposes to measure the curvature of neural networks by dividing the spectral norm of the input-output Hessian by the L2 norm of the gradient. It argues that this normalized curvature measure has the advantage of being invariant to gradient rescaling. The paper then introduces an approach for training low-curvature networks by using a modified softplus nonlinearity and Lipschitz-constrained linear and batchnorm layers and demonstrates that networks trained in this manner are more robust to adversarial attacks. It also observes that the proposed curvature measure is significantly more consistent between the train and test set than the gradient and Hessian norm.

**Questions:**

* Please provide some discussion around flat regions (see limitations). It may not be an issue, but the addition of the small constant suggests to me that you are aware that there might be a problem .
* Could GradReg be combined with CURE? If so, this should be included as a baseline to have a fair comparison with your LCNNs + GradReg.
* My main suggestion would be to either focus on establishing the proposed curvature measure as a generally useful tool or broaden/deepen the empirical comparison. At the moment the paper sits in between two chairs in my eyes and does not make a convincing case on either of them.

Minor:
* I find the title to be of somewhat questionable taste given that many readers (if the paper is accepted) will have been affected by the pandemic and some may have strong opinions on covid-related policies one way or another (I will caveat this with saying that I dislike click-bait titles and prefer them to be informative). I would encourage the authors to drop the "flatten the curve" bit.
* I would suggest clarifying in the abstract that you are considering input-output Hessians/gradients (when I read gradient/Hessian I think of the loss w.r.t. the parameters, which surely is just my personal bias, but that may be shared by other readers).

Note regarding the score: I would have put this as a borderline reject in past years, but feel comfortable erring on the lower side since that score seems to be discouraged this year.

**Limitations:**

The paper somewhat sweeps the (potential) issue of the curvature measure not being defined in zero-gradient regions of the input space under the rug by adding "a small constant to ensure numerical stability" to the denominator. This is a fairly nonsensical justification since for a symbolic definition numerical stability is not a concern until the measure is implemented. I would strongly suggest adding an explicit discussion rather than side-stepping the issue in this manner. Clearly, for networks with a relu nonlinearity after the input layer, such flat regions in the input space exist, although this may not be the case for the suggested centered softplus nonlinearities.

**Strengths And Weaknesses:**

Strengths:
* The paper is clearly structured and easy to follow.
* The (overall) approach is novel as far as I am aware (although see weaknesses on the Lipschitz batchnorm layers).
* General purpose parameterization/regularization approaches as presented here are potentially useful beyond adversarial examples.

Weaknesses:
* The proposed curvature measure is not (really) defined in zero-gradient regions and there is no discussion around this (see limitations).
* The motivation of the curvature measure is fairly hand-wavy -- do we want to learn (approximately) locally linear functions when training neural networks? This might be the case, but would need to be supported with references. Moving up section 4 (4.2 in particular) could help, but would make this more of an adversarial robustness paper (which might help clarifying the storyline, see next point).
* I find the overall storyline of the paper a bit incoherent, it spends a fair amount of space justifying the curvature measure, but is ultimately not really theoretically focused. On the other hand, the empirical evaluation is fairly limited (just one basic setting that isn't just for illustration) and does not make a particularly strong case for the proposed method. I'd suggest either expanding the theoretical discussion around the curvature measure or expanding the experimental evaluation and demonstrating the benefit of low-curvature networks in other use cases than adversarial examples or compare to SOTA attacks/defences (I don't follow the literature on adversarial robustness, but my impression is that this has been a fast-moving field over recent years, so comparing on a single attack type to effectively two baselines seems quite limited).
* If I'm not mistaken, the Lipschitz parameterization of batchnorm layer has previously been proposed in (Gouk et al., Regularisation of Neural Networks by Enforcing Lipschitz Continuity, arXiv:1804.04368, 2018).
* The proposed network parametrization appears to not work particularly well on its own and relies on the simultaneous use of gradient norm regularization (at least for the adversarial robustness use case).

---

> ### Author Response · Authors · 2022-08-02
> **Response to reviewer questions**
>
> Thank you for your in-depth review, we really appreciate your insightful comments. We respond to your questions below.
>
> 1.  **Curvature is ill-defined at flat regions; numerical stability is not an appropriate justification**:  Thank you for bringing this up - this is indeed a valid point! For small values $\varepsilon$ and for points where gradients are close to zero, there is indeed a risk of the curvature growing to be large.
>
>     First, despite the above point, it is clear technically from definition that the proposed curvature measure is well-defined at all points, and the addition of the positive scalar $\varepsilon$ ensures this. However we agree that the term ‘numerical stability’ might not be appropriate to describe this, and we will remove this term from the paper. The motivation to add this term is to ensure that the curvature measure is well-defined everywhere.
>
>     Second, for any $\varepsilon$, the data-free upper bounds in Thm 1 still holds, as the addition of $\varepsilon$ can only decrease the curvature value and in particular is always well-defined. Our bound in Section 4.1 can also be appropriately tweaked by adding a similar $\varepsilon$ in the denominator.
>
>     Third, we never noticed any stability issues practically in our experiments, either during training low curvature models or while empirically measuring model curvature for Table 1. We hypothesize that this is due to the difficulty of finding examples $x$ where the model gradients are vanishingly small compared to the Hessians. This may be due to the fact that the set of such local optima are extremely sparsely located in the input space.
>
>     We thank you again for bringing up this issue, and we will thus add this discussion to the paper.
>
>
>
> 2.  **Motivation of the curvature measure is hand-wavy**: We apologize if the motivation was unclear. We want to train low curvature models because (1) they have stable gradients (Sec. 4.1), (2) they are necessary for robustness (Sec. 4.2), (3) Occam’s razor and avoiding overfitting (e.g.: Figure 1). We will move Section 4 earlier in the paper as you mention.
>
> 3. **Incoherent storyline**: We are sorry that it came across this way! Our intended storyline is as follows - our objective is to introduce efficient tools for training low-curvature models, and our experiments show that we are indeed effective in doing so (Table 1). We show theoretically (Sec. 4) and empirically (Figure 2 and Table 2) that these have benefits in providing stable gradients and $\ell_2$ robustness.
>
>     With regards to your comment on "demonstrating the benefit of low-curvature networks in other use cases than adversarial examples", stable gradients are desirable for interpretability of gradients as saliency maps as well (https://arxiv.org/abs/2203.06877), where they are a metric to measure the quality of a proposed gradient map. Thus we argue that low curvature models are useful for gradient interpretability as well.
>
>     We do not aim to be an adversarial robustness paper, however we believe it is a **surprising result** that LCNNs out of the box can compete with PGD trained models. As requested by several reviewers, we also run new experimental results on auto-attack, and observe that our claims do not change. We present these results in the comment common to all reviewers.
>
> 4. **Lipschitz parameterization of batchnorm layer has previously been proposed**: Thank you for pointing us to this paper! We were unaware of this paper, and find that it indeed proposes something similar to our $\gamma$-Lipschitz batchnorm. The primary difference seems to be that while we fit each batchnorm layer to a different $\gamma$ value, Gouk et al have all batchnorm layers in a model have the same $\gamma$ value, thus restricting flexibility.
>
>    However note that this is not one of the major contributions of our paper, and we believe $\gamma$-BN is a rather natural solution to the problem of having layers with bounded Lipschitz constants. Nonetheless, we will add a discussion about this in our paper.
>
> 5. "relies on the simultaneous use of gradient norm regularization (at least for the adversarial robustness use case)" : Please note that this is expected even in theory, from Section 4.2, where find that both low curvature and low gradients are necessary ingredients of robustness. This is in fact a salient feature of our curvature measure, which explicitly disentangles the second order effects (“how linear is my model”) from first order ones (“how steep is my model locally”), which does not happen with previous notions of curvature used in literature.
>
>
> (Responses continued in the next comment)

---

> > ### Author Response · Authors · 2022-08-02
> > **Response (Continued)**
> >
> > (Continued from the previous comment)
> >
> > 6. **CURE + GradReg**: We train a model with CURE + GradReg, and measure its $\ell_2$ PGD robustness below and compare with LCNN + GradReg as requested. We find that including gradient norm regularization further degrades the clean accuracy in favour of adversarial robustness. We believe it may be possible to tune the hyper-parameters of CURE + GradReg to obtain a more favourable clean accuracy vs robust accuracy trade-off, upon further experimentation.
> >
> >     The table below shows accuracy numbers similar to Table 2 in the main paper.
> >
> > | Model          | eps=0.05 | eps=0.1 | eps=0.15 | eps=0.2 | Clean acc |
> > | -------------- | -------- | ------- | -------- | ------- | --------- |
> > | LCNN + GradReg | 72.68    | 63.36   | 52.96    | 42.7    | 77.29     |
> > | CURE + GradReg | 73.30    | 68.80   | 63.30    | 56.55   | 74.79     |
> > | CURE           | 71.39    | 61.28   | 49.60    | 39.04   | 76.48     |
> >
> > ---
> > Minor comments
> >
> > "title to be of somewhat questionable taste": We did not have an intention to offend with this title. We shall drop the offending portion in our update.
> >
> > "clarifying in the abstract that you are considering input-output Hessians/gradients": This is a good point, and we will do so!
> >
> > ---
> >
> > We would be happy to answer more questions, and if you believe we have addressed some of your concerns adequately, we would ask you to consider raising your score. Thank you!

---

> > ### Comment · Reviewer_1uxf · 2022-08-06
> > **Thank you for the response**
> >
> > Dear authors, thank you for the detailed response, I found it really constructive and factual.
> >
> > I appreciate the additional results with CURE + GradReg as well as the evaluation on another type of adversarial attack in addition to the ablation study. I think this strengthens the empirical contribution of the paper which had previously been a weakness, so I will increase my score. I am still undecided whether I want to recommend acceptance, I will reconsider this after the reviewer discussion.
> >
> > Finally, whether the paper is accepted or needs to be resubmitted, I think an empirical demonstration of the curvature measure/regularization helping beyond adversarial robustness would greatly strengthen the paper, convincingly broadening its scope beyond that of an adversarial defense paper.

---

### Official Review · Reviewer_J6Z1 · 2022-07-11

**Rating:** 4
**Confidence:** 3
**Soundness:** 3 good
**Presentation:** 3 good
**Contribution:** 2 fair

**Summary:**

The authors propose to build a kind of low-curvature neural networks (LCNNs), which can lower network curvature for achieving robust models. The key to reducing the curvature in LCNNs is centered-softplus activation function and $\gamma$-Lipschitz BN layer, which are specifically designed by the authors with better curvature property compared to those standard layers. Meanwhile, the authors give a theoretical upper bound on the curvature of networks, and demonstrate that these layers can minimize such bound. Finally, the authors empirically show that LCNNs can give lower curvature via training Resnet18 on Cifar100, and also show that LCNNs can improve the performance further in combination with the GradReg in adversarial experiments.


**Questions:**

I am willing to raise my score if the authors can address my concerns.

**Limitations:**

In my opinion, this paper do not involve related issues for its current version.

**Strengths And Weaknesses:**

&nbsp;
## **Strengths**:
1. The paper is clear and well written.
2. The authors show some interesting insight about the curvature property of the stacked layer networks.
3. The proposed centered-softplus has some nice properties, and can be promising to be implemented in future model.
4. The LCNN models show a universal improvement in adversarial experiments.


&nbsp;
## **Weaknesses**:

*Q1*:

The authors claims that conventional measures such as Hessian norm is sensitive to gradient scaling (section 3.1). And the authors propose a kind of definition of curvature that is approximately normalized, $C_f(x) = ||\nabla_x^2 f(x)||_2 / (||\nabla_x f(x)||_2 + \epsilon)$. As far as I know, Dinh (citation 1 in the authors' paper) also gives a definition that is very similar to the authors', which is $|\nabla^2 L(\theta)||_2\epsilon^2 / 2(1 + L(\theta))$ (Left column at page 3 in their paper). Note that Dinh's form is to measure the curvature in the weight space while the authors measure in the input space, but this does not affect the measure metric. So I am wondering what is the main difference between the two formula in essence.

*Q2*:

The authors try to change the function property to reduce the network curvature, where the authors propose the centered-softplus activation function. My second concern is that to my understanding, the low curvature will normally denote a low gradient norm. The Lipchitz constant of a function equals to the maximum gradient norm for a small region. A low curvature generally means that the change in the neighborhood is relatively small, which implies a low Lipchitz constant and then a small gradient norm. So in our training, we will use the typical backpropagation to sequentially update the weight at each layer according to the gradient of pervious layers. My concern is that if a function has a low curvature, we may more likely to suffer from the gradient vanishment for these shallow layers far from the output layer. This is the reason why the relu function is mostly used for building models. Considering that the c-softplus is a low curvature function, especially around zero, I am somehow worried about the side-effect it brings in training, i.e. the gradient vanishment. Also, in todays modeling concept, the community are building very deep models. I think that the techniques proposed by the authors may not be well suited for these scalable deep models. Low curvature may decrease their expressivity, since the weights in the shallow layers are limited to some extent. The authors only use ResNet18 for demonstration, so it will be more interesting to see such layer to be applied in deeper models.

*Q3*:

My third concern is about some details in experiments. Firstly, since training LCNN will further involve two extra hyper-parameters, so I am wondering how the authors tune these parameters in practice. Is there any advice about the two parameters? Based on the experiment results, introducing two extra parameters can only improve little performance, but will severely burden the tuning. Secondly, to my understand, $\nabla^2 f$ will not be computed directly, so I am also wondering how this is computed. Thirdly, the computation cost of training with the authors' scheme is 1.6 times higher than that using the standard scheme. So in my opinion, such an additional cost is not quite proper to be called as "marginally increasing the training time".

#### **Minor**:

It will be very interesting to see that the authors train the LCNN with SAM (Sharpness-Aware Minimization for Efficiently Improving Generalization, Pierre Foret et, ICLR2021), which is a techniques used for smoothing the loss curvature.

---

> ### Author Response · Authors · 2022-08-02
> **Response to reviewer questions**
>
> Thank you for your review, we highly appreciate your comments. Please find below our responses to your questions.
>
>
> 1.  **Difference w.r.t. curvature measure used by Dinh et al**: While it is true that both expressions look similar on the surface, Dinh et al scale the Hessian with the loss value itself, whereas we scale it with the gradient. Thus while the curvature measure of Dinh et al is roughly invariant to function scaling, whereas we are invariant to both function scaling and gradient scaling. In addition, they add a value of one to the loss in the denominator, whereas we add a generic value of $\epsilon$.
>
>
> 2.  “.. low curvature will normally denote a low gradient norm..”: This is incorrect, and this is precisely what we aim to disentangle in our paper. As an example, a deep **linear** model has zero curvature, and yet does not have vanishing gradients. Thus low curvature is unrelated to low gradient norm. Experimentally, in Table 1, we can compare the gradient and curvature values for "LCNN" and "GradReg". While LCNN has smaller curvature and larger gradient norm in that table, GradReg also larger curvature and smaller gradient norm. This is in fact a salient feature of our curvature measure, which explicitly disentangles the second order effects (“how linear is my model”) from first order ones (“how steep is my model locally”), which does not happen with previous notions of curvature used in literature.
>
>
> 3.  **Tuning hyperparameters**: We chose hyper-parameters ($\lambda_\beta,\lambda_\gamma$) from the set $[10^{-1}, 10^{-2}, 10^{-3}, 10^{-4}, 10^{-5}, 10^{-6}]$ and chose the largest hyper-parameters that still resulted in no drop in training accuracy.
>
>     **Hessian norm computation**: We do not need to compute the Hessian to train our models. For table 1, we use Pearlmutter’s trick [1] to compute Hessian vector products, and use the power method on top of this to compute the largest eigenvector, which is the definition of the spectral norm. We will mention this in the paper.
>       [1]: Fast Exact Multiplication By the Hessian; Pearlmutter, 1993
>
>     **Claims of marginal increase overblown**: We apologize for the usage of the phrase "marginally increasing the training time" in reference to our method. We will remove this phrase in the paper. Our phrasing in the paper was to point to the fact that our methods represent a marginal increase in training time *compared* to alternatives such as CURE or Adversarial training, which take 5x as much time.
>
> 4.  **Comparison with SAM**: We believe that the objective of SAM is orthogonal to our goals - while we are interested in low curvature in the input space, SAM attempts to find weights that are locally flat in the weight space.  Thus while we believe this exploration is interesting, this may be outside the scope of this rebuttal.
>
>
> We would be happy to answer more questions, and if you believe we have addressed your concerns adequately, we would ask you to consider raising your score. Thank you!

---

> > ### Comment · Reviewer_J6Z1 · 2022-08-07
> > **Thank the author for the response.**
> >
> > 1. I massively appreciate the authors' interpretation. But I can tell these differences replied by the authors from the formula. So my question is if there is any essential difference between them. For example, whether the absolute value of the loss itself matters in assessing the curvature in such scenario.
> >
> > 2. Thank the authors for the response. Firstly, I have interpreted that to some extent that the curvature is associated with the gradient, but definitely not in an absolute manner. This is partly because that low curvature intuitively means that the values in the neighborhood are close, and gradient would give a description about the variation in the neighborhood. And we are focusing on the deep models, which is heavily non-convex and complex. Even for a deep linear model, it would not definitely have a zero curvature. For example, a simple two layer linear model (i.e. without non-linear activation), which functions as $y = x^2$, I am quite sure that this model could not have zero curvature everywhere. So I am also wondering what kind of linear model will have zero curvature in the authors' example. Secondly, in my opinion, all these interpretation by the authors would not be more convincing than simply showing a successful training on deeper models, even just the ResNet architecture. In summary, in my opinion, all the authors' response have not shown well that the proposed methods would not suffer from the possible side-effect of gradient vanishment, especially for scalable models.

---

> > > ### Author Response · Authors · 2022-08-08
> > > **Clarification regarding Curvature**
> > >
> > > Thank you for your response! We clarify these points below.
> > >
> > > (1) As we mention in our rebuttal, while the curvature measure of Dinh et al is roughly invariant to function scaling, whereas we are invariant to both function scaling and gradient scaling. In other words, if the loss at a point is scaled by a large value $k > 0$, then so is the Hessian, leaving the Dinh et al curvature measure unchanged wrt scaling of the loss function. In our case, either scaling the function or the gradients leaves our proposed curvature measure unchanged. We discuss this property in lines 101-106 in the main paper as it relates to our method. If your question pertains to some other property of this curvature measure, please let us know, we would be happy to discuss this.
> > >
> > > (2) **"low curvature intuitively means that the values in the neighborhood are close"**: This is incorrect when applied to our proposed curvature measure. If the curvature is defined as simply the Hessian, then local flatness (small gradients) may imply having a small Hessian and thus a small curvature. However this is not so for our curvature measure, which explicitly divides Hessian by the gradient. For our case, low curvature intuitively denotes how linear the underlying model is. For gaining such intuition, consider a finite-difference perspective, where the Hessian norm is given by the difference of gradients of nearby points. This is small when either the gradients themselves are small, or nearby gradients are similar to each other. By dividing by the gradient itself, we ensure that when our curvature measure is small, it is always because nearby gradients are similar to each other, and not because the function is locally flat.
> > >
> > > Note that this is broadly a property of most curvature measures, not just ours. Quoting from https://en.wikipedia.org/wiki/Curvature: "Intuitively, the curvature is the amount by which a curve deviates from being a straight line, or a surface deviates from being a plane", and "The curvature of a straight line is zero".
> > >
> > > **"a simple two layer linear model (i.e. without non-linear activation), which functions as $y=x^2$, I am quite sure that this model could not have zero curvature everywhere"**: Please note, $y=x^2$ is **not** a linear model. A deep linear model is instead defined as a composition of linear layers, which results in a linear model overall, whereas $y = x^2$ is non-linear. For any linear model (assume one-dimensional function for clarity here) $y = Wx + b$, the gradient w.r.t. input is equal to $\frac{dy}{dx} = W$, and the Hessian is zero ($\frac{d^2y}{dx^2} = 0$). Thus the curvature of linear models is always zero. Also note that if $||W||$ is large, this model can have large unbounded gradients, but still zero curvature.
> > >
> > >
> > >
> > > **"all the authors' response have not shown well that the proposed methods would not suffer from the possible side-effect of gradient vanishment, especially for scalable models."**: Apologies for not being clear about this. Our arguments above were intended to show analytically that there is no connection between having low curvature (model being locally linear) and low gradient norms (model being locally flat) and thus vanishing gradients. Our method aims at training models that are "as linear as possible" (i.e., definition of low curvature), and thus this is unrelated to having small gradients.
> > >
> > > Empirically, while training ResNet-18 architectures on CIFAR-10 / CIFAR-100, we did not find any training instabilities, and in particular we used the same standard training hyper-parameters for all our architectures (line 269 in our paper). Also please note that the **gradient norms** of the LCNN model in Table 1 are on par with (or even higher than) gradient norms of standard models, showing that there is **no vanishing gradient effect** with LCNNs.
> > >
> > > We shall update our comment soon with some more preliminary empirical results on ResNet-34 on CIFAR-10/100.
> > >
> > > In the meanwhile, please let us know if you have any further questions regarding the points made above. Thank you again for actively engaging with us!
> > >
> > > ---
> > >
> > > **Update**: We present below preliminary empirical results on ResNet-34 on CIFAR-10 and CIFAR-100, trained by using the same hyper-parameters in the ResNet-18 case. In both cases, we observe that we are able to train models successfully, and we see gradient norms of LCNNs being similar to that of standard models, thus showing **a lack of vanishing gradients**. Note also that LCNNs have drastically lower curvature than standard models as usual.
> > >
> > >
> > > **Results on CIFAR-10:**
> > >
> > > | Model | Curvature | GradNorm | Hessian | Accuracy |
> > > | ---  | ---  | --- | --- | --- |
> > > | Standard | 282.20   | 9.6      | 2905.5  | 94.00%   |
> > > | LCNN  | 55.73     | 7.6      | 439.4   | 93.86%   |
> > >
> > > **Results on CIFAR-100:**
> > >
> > > | Model | Curvature | GradNorm | Hessian | Accuracy |
> > > | ---  | --- | --- | --- | --- |
> > > |Standard| 265.23   | 26.24    | 7325.38 | 76.23%   |
> > > | LCNN  | 78.82     | 31.92    | 2171.25 | 77.43%   |

---

> > > > ### Comment · Reviewer_J6Z1 · 2022-08-10
> > > > **Thanks for the author's response.**
> > > >
> > > > We appreciate the author for the kind response. I would like to discuss more about the reponses.
> > > >
> > > > (1) "if the loss at a point is scaled by a large value", I am wondering why and in which case would we need to be invariant to such scaling? In my opinion, the definiation of curvature should focus more on the rescaling operation introduced in Dinh's paper, which functions on the weights and will not affact the results.
> > > >
> > > > (2) "This is incorrect when applied to our proposed curvature measure." Firstly, if the author would like to argure about "low curvature intuitively means that the values in the neighborhood are close", I think the best way is to provide a counter example where "the values in the neighborhood are not close but the curvature is low" in an intuitive way. Note that I said in my comment "it intuitively means" here, so not in a deterministic manner. Secondly, since the author also could not theoretically prove that the proposed the metric could measure the curvature in an accurate way, if the author could not give a counter example, I have to say that it just proves in some way the proposed curvature measure is incorrect.
> > > >
> > > > In my opinion, the curvature is a rather complicated concept, especially for such huge dimensional manifolds. Although some scalar indicators can somehow be used as a overall description of flatness, yet it is still too rough to fully characterize the curvature of the manifolds in each dimension. Many scalar metrics may be meaningful in some perspectives and not be not meaningful in other perspectives. I admit the author's metric make sense in some situations, but it could definitely not be able to fully describe the flatness in high dimention. So is the gradient norm.
> > > >
> > > > About $y = x^2$, sorry, my mistake. I have writen not correctly and gave the wrong example. My point is that the optimizations of linear models and non-linear ones are different, where non-linear models are much more complicated, which is why we investigate curvature. Meanwhile, a successful training on linear models do not mean it could achieve zero training error like non-linear ones i.e. the results are satisfactory .
> > > >
> > > > I think my concern still exists. Gradient vanishment is mainly happened to shallow layers, like I said in the original comments. The author have not discussed this point. The gradient norm of the whole network would not be helpful.
> > > >
> > > >  And by the way, I could not get catch the core difference between the locally linear and locally flat, I am wondering could the author give some clear definition about what is locally linear and what is locally flat?
> > > >
> > > > I appreciate the additional results very much. It relieves my concerns to some extent.  However, it is still not scalable models in today's training art.
> > > >
> > > > And I would appreciate the author's further response so I could re-estimate the score.

---

### Official Review · Reviewer_o9LW · 2022-07-11

**Rating:** 6
**Confidence:** 5
**Soundness:** 3 good
**Presentation:** 3 good
**Contribution:** 2 fair

**Summary:**

This paper proposes a set of modifications to neural networks to control their curvature. Three modifications are proposed: 1) Centered-softplus activation functions 2) Spectrally normalization of fully-connected, convolutional, batch normalization layers 3) regularization of activation and batch normalization learnable curvatures parameters. In particular, the paper proposes an alternative measure of curvature that is inversely proportional to the norm of the gradient. The paper evaluates curvature controlled models for robustness.

**Questions:**

- Have you evaluated the method when combined with adversarial training?
- ​​Eq 4: Can you explain why softplus converges to x/2 when beta goes to zero?

**Limitations:**

Yes

**Strengths And Weaknesses:**

Strengths:
- A curvature measure coupling both the Hessian and gradient is interesting. Because of division by the gradient norm, the method might learn to automatically stay away from gradient masking and zero gradients.  Table 1 supports this by showing that the norm of both the gradient and the Hessian are smaller than other robustness methods considered.
- Centered softplus is also an interesting modification to softplus as it ensures increasing the depth does not result in activation blowup near zero.

Weaknesses:
- There have been challenges with second-order approaches to robustness such as gradient stability and gradient masking [2]. Although the ideas in this work could resolve issues in prior work, we do not see evaluations of gradient stability and gradient masking. For example, plots of the gradient norm during the training of various models on various datasets would show us if the curvature regularization is actually stable and allows us to train any model architecture. More importantly, the paper should test the model for gradient masking using for example black-box attacks. Adaptive attack should also be considered. For example, make sure there is no spectral normalization at test time by substituting the weights with their normalized copy and replace softplus with a piecewise linear approximation such as relu at test time.
- Robust evaluation does not include strong attacks such as the ones in the RobustBench [2]. The strongest attack is PGD and the number of steps is not mentioned. Also, the evaluation is limited to L2 robustness without a discussion on why the method may or may not work for Linf robustness.
- Ablation studies are required to disentangle the impact of three proposed modifications.
- Related works such as [1,3,4] are missing from comparison and discussion.

[1] Ma, A., Faghri, F., Papernot, N., & Farahmand, A. M. (2020). Soar: Second-order adversarial regularization. arXiv preprint arXiv:2004.01832.
[2] Croce, F., Andriushchenko, M., Sehwag, V., Debenedetti, E., Flammarion, N., Chiang, M., ... & Hein, M. (2020). Robustbench: a standardized adversarial robustness benchmark. arXiv preprint arXiv:2010.09670.
[3] Anil, C., Lucas, J., & Grosse, R. (2019, May). Sorting out Lipschitz function approximation. In International Conference on Machine Learning (pp. 291-301). PMLR.
[4] Qin, C., Martens, J., Gowal, S., Krishnan, D., Dvijotham, K., Fawzi, A., ... & Kohli, P. (2019). Adversarial robustness through local linearization. Advances in Neural Information Processing Systems, 32.

---

> ### Author Response · Authors · 2022-08-02
> **Response to reviewer questions**
>
> Thank you for your review, we highly appreciate your comments!
>
> Before proceeding, we would like to address a misunderstanding with regards to the goals of our paper. Our objective in this paper is to propose tools to build low-curvature neural networks. Our objective is **not** to obtain SOTA results on adversarial robustness, although we find that LCNNs **out of the box** are competitive with PGD-trained models, which is a surprising result to us. Having said that, we understand that more evaluations may be necessary to evaluate the utility of our approach, and provide some such results in the comment addressed to all reviewers.
>
> 1.  **Evaluations of gradient stability**: Sections 4.1 and 5.2 evaluate gradient stability / gradient masking both theoretically and experimentally. These results show that our curvature regularizations **increase** gradient stability and thus **decrease** gradient masking, by making the model gradients smooth and well-behaved everywhere. On the other hand, gradients of standard ReLU models are not well-defined at the ReLU kinks, and neither are they guaranteed to be smooth. We also provide results on auto-attack in a comment addressed to all reviewers, which contains the "square" black-box adversarial attack.
>
>
> 2.  **More robustness evaluations**: Our threat model is that of $\ell_2$ white-box adversarial robustness in line with our theoretical results in Section 4.2. We run PGD for 10 iterations to produce results in Table 2, and we apologize for omitting this detail. We also compare against auto-attack in a comment addressed to all reviewers, to address the comment.
>
>
> 3.  **Ablation studies**: Please find the ablation studies in a comment addressed to all reviewers, to address this comment.
>
>
> 4.  **Related works discussion**: Thank you for suggesting these references, we will discuss these in our paper.
>
> 5.  "Have you evaluated the method when combined with adversarial training?" : We are currently running this method, and will provide an update once we complete the experiment.
>
> Update: we have completed this experiment, and present results in the global comment addressed to all reviewers. We find that combining LCNN + GradReg (our method) with adversarial training results in further increase of robust accuracy, but at the cost of clean accuracy.
>
> 6. "Can you explain why softplus converges to x/2 when beta goes to zero?" : We can see this by applying L'Hopital's rule on equation 4.
>
>
> We would be happy to address any remaining concerns, and if you believe we have addressed your concerns adequately, we would ask you to consider raising your score.
>
> Thank you!

---

> > ### Comment · Reviewer_o9LW · 2022-08-07
> > **Thank you for your response**
> >
> > Most of my concerns have been addressed through the added experiments. In particular, the results for the mixture of the proposed method and adversarial training are very promising. The gradient masking evaluation through a black box attack further strengthens the paper. I would still be interested in my suggested experiment of an adaptive attack. But I'm convinced to raise my score.

---

### Official Review · Reviewer_b1TK · 2022-07-12

**Rating:** 7
**Confidence:** 4
**Soundness:** 3 good
**Presentation:** 3 good
**Contribution:** 3 good

**Summary:**

The authors provide a new method for minimizing decision boundary curvature in an effort to improve adversarial robustness without compromising unperturbed generalization (test set) accuracy. They propose combining an activation function with a trainable curvature parameter as well as a Lipschitz constrained batch-normalization layer. Their empirical results demonstrate success in their goal of maintaining accuracy while improving robustness.

**Questions:**

1) Can you compare your definition of curvature to that found in differential geometry? The concept of curvature in n-dimensional spaces has been long established in the differential geometry community, and is associated with the process of Principal Curvature Decomposition. The most accessible work on the subject, in my opinion, is from John M Lee [1, 2]. Importantly, following Lee, the curvature of a Riemannian submanifold (which a NN decision boundary reasonably approximates) must be calculated by projecting the Hessian onto the function’s level-set (i.e. decision boundary) and then normalizing by the first fundamental form, or metric (see [2], chapter 8). This is very close to what you propose, but it is different in that your proposal is missing the projection operation. Moosavi-Dezfooli also mentions this in [3], section 5, although I believe they are also missing details found in the more formal treatment provided by Lee. For additional context, another similar treatment can be found in [4].  I think it would improve the quality of the submission if the authors provided some context for how their work fits into this broader mathematical description of curvature.

2) What is the relationship between decision boundary curvature and neural activation curvature? From my understanding, the prior work from Moosavi-Dezfooli attempts to minimize a network’s decision boundary curvature directly. On the other hand, your regularization on the beta term in the centered softplus activation function is intended to encourage more linear (i.e. flatter) activation functions for each neuron in all layers and the batch-norm regularization is layer specific. Following Moosavi-Dezfooli and colleagues, for some logit (i.e. penultimate layer) neurons f_i(x), f_j(x), the binary decision boundary can be written down as a b_{ij}(x) = f_i(x) - f_j(x). So one can pretty easily deduce how the curvature of f_i and f_j impacts b_{ij}. However, the relationship between neuron activation curvature and decision boundary curvature for earlier layers is not clear to me. It seems reasonable to think that individual neurons could exhibit curvature in their activation maps while still contributing to (relatively) flat decision boundaries downstream. Conversely, ReLU networks are piecewise linear, and therefore technically contribute zero curvature in the decision boundary space. Even if we squint and allow for an approximate definition of curvature (in the sense that piecewise linear functions approximate a curve in the limit), we have an example of a linear-almost-everywhere activation function producing non-flat decision boundaries. Figure 1 and table 1 both seem to indicate that the proposed regularization is indeed minimizing boundary curvature, but I am interested to hear from the authors why that is happening in their case.

3) Is it possible to do more ablation studies? My apologies if I missed this, but is there an analysis of the individual effects of the new activation function vs the batch norm modification vs the combination?

4) You may consider a more thorough robustness evaluation. As I’m sure the authors are aware, it is difficult to validate adversarial robustness claims, and many studies that initially claim robustness ultimately prove to fail under more careful scrutiny [5, 6]. While I think it is unreasonable to ask the authors to satisfy every consideration made in [5], I do think additional care should be taken (e.g. providing results for l1 bounded attacks, for gradient-free attacks, and/or auto-PGD attack). I would also make sure to include a section (appendix would be fine) that explains the thread model and what care was taken to explore the parameter space of the attacks.

5) typo on line 231: “that the function [is] locally quadratic”

[1] John M. Lee. Introduction to Smooth Manifolds. Number 218 in Graduate Texts in
Mathematics. Springer, 2 edition, 2013. ISBN 978-1-4419-9981-8 978-1-4419-9982-5.

[2] John M. Lee. Introduction to Riemannian Manifolds. Springer International Publishing, 2nd edition, 2018. ISBN 978-3-319-91754-2.

[3] Seyed-Mohsen Moosavi-Dezfooli, Alhussein Fawzi, Omar Fawzi, Pascal Frossard, and Stefano Soatto. Robustness of classifiers to universal perturbations: A geometric perspective. In International Conference on Learning Representations, ICLR, 2018.

[4] Ben Poole, Subhaneil Lahiri, Maithra Raghu, Jascha Sohl-Dickstein, and Surya Ganguli. Exponential expressivity in deep neural networks through transient chaos. Advances in neural information processing systems, 29:3360–3368, 2016.

[5] Nicholas Carlini, Anish Athalye, Nicolas Papernot, Wieland Brendel, Jonas Rauber, Dimitris Tsipras, Ian Goodfellow, Aleksander Madry, and Alexey Kurakin. On evaluating adversarial robustness. ArXiv preprint, abs/1902.06705, 2019. URL https://arxiv.org/abs/1902.06705.

[6] Croce, Francesco, and Matthias Hein. "Reliable evaluation of adversarial robustness with an ensemble of diverse parameter-free attacks." International conference on machine learning. PMLR, 2020.


**Limitations:**

The authors provide a list of limitations in the discussion.

**Strengths And Weaknesses:**

**Originality:** As the authors state, the concept of adapting boundary curvature to improve robustness is not new. Additionally, as I mention in point 1 below, the idea of normalizing the network hessian by the gradient is also not new. However, as far as I know their method of using an activation function with a learnable curvature parameter coupled with the bounded batch normalization layer is indeed novel. I believe this study provides an interesting (and original) piece of evidence supporting a budding theory about the relationship between decision boundary curvature and adversarial robustness.

**Quality:** The figures illustrate their intended points well. I did not attempt to replicate or validate any of the mathematics, but I did read it for correctness and found no mistakes. However, I have requests below for more rigor in the experimentation and mathematical exposition.

**Clarity:** Overall the paper is clearly communicated.

**Significance:** While the ideas presented are not too far from existing work, I still think the paper is significant. Adversarial training continues to be the only long-standing applicable defense against adversarial attacks, and demonstrates a widely-believed tradeoff between accuracy and robustness. In my opinion, the significance of this submission is that their results suggest an alternative method which defies the accuracy-robustness tradeoff. However, I am assuming that their measurements of curvature and robustness are correct, and I make a couple of notes below asking for more support to ensure that this is true.

---

> ### Author Response · Authors · 2022-08-02
> **Thank you for you review!**
>
> Thank you for your in-depth review and insightful comments. We really appreciate it.
>
> Before proceeding, we would like to address a minor mis-understanding: the objective of our paper is not to minimize the curvature of the **decision boundary** but that of the **overall input-output mapping** of the function itself. However, as you mention in your review (point 2), low curvature input-output maps do imply low curvature decision boundaries, and we also see this experimentally in our toy example in Figure 1, and further discuss this issue in point 2 below.
>
> We now begin to address your specific questions.
>
> 1.  **Comparing proposed definition of curvature to standard definitions in differential geometry**: This is an interesting question. We hadn’t considered analogous definitions from differential geometry, mostly because our starting point was other work equating the curvature of functions with Hessian norms (e.g. Dombrowski et al, 2022; Moosavi-Dezfooli et al, 2018; Dinh et al, 2017).
>
>     Unfortunately, we do not see a straightforward connection between the differential geometric definition of curvature and ours. This is mostly because the definition you mention in your review, i.e., "(curvature) ... must be calculated by projecting the Hessian onto the function’s level-set" is not applicable in our case as we are interested in the measuring the curvature of the input-output map and not its decision boundary. However we believe exploring these alternate definitions from differential geometry is an interesting problem.
>
>
> 2.  **Impact of function curvature on decision boundary**: We can argue that for any function $g$ composed with a low curvature function $f$, the function $g \circ f$ subsequently has low curvature, provided that the $g$ itself has bounded curvature and Lipschitz constants. Thus the function that takes the model logits $f$ to the decision boundary can be seen as one such function $g$, leading the decision boundary $g \circ f$ to have low curvature. This is similar to the reasoning done in Section 2 of the supplementary material, where we connect loss and logit curvatures.
>
> 3.  **Ablation studies**: We present results of ablation studies in the comment addressed to all reviewers.
>
> 4.  **Robustness experiments**: We present results on additional robustness experiments on auto-attack in the comment addressed to all reviewers.
>
>     Throughout the paper, our primary threat model is that of $\ell_2$ white-box adversarial attacks, as our bounds in section 4 all involve $\ell_2$ distances. We ran PGD for 10 iterations to generate our results in Table 2.
>
>
> We would be happy to answer more questions, and if you believe we have addressed your concerns adequately, we would ask you to consider raising your score. Thank you!

---

> > ### Comment · Reviewer_b1TK · 2022-08-10
> > **Response to rebuttal**
> >
> > Thank you for the response and especially for the additional experiments. I believe the added ablation & robustness studies will improve doubts from skeptical readers and the general defensibility of the paper. Overall I think the study is a valuable contribution, and I will improve my score to reflect this.
> >
> > Regarding the curvature definitions:
> >
> > You're right to point out the distinction between decision boundary curvature (which can be defined as a level set) and function curvature. I do believe these concepts can be related pretty easily, and ultimately that relationship might come down to simply using the gradient normalized Hessian, as you have done. But I don't know exactly how that would look, and I don't think it is something you should have to do for this study. Maybe consider it for future work.
> >
> > However, I recommend that you make the distinction clear in your paper, since in fig 1 you're showing decision boundaries, which might confuse a reader given the body of literature (e.g. from Moosavi-Dezfooli and colleagues) looking at boundary curvature.
> >
> > I also recommend that you tone down the claim of novelty for contribution #1. I agree that my example of measuring boundary curvature is a different application from yours, and thus the work in that area from e.g. Moosavi-Dezfooli doesn't necessarily count as prior work. However, the concept of scaling the hessian by the gradient is the literal definition of function curvature, as is described in Lee, which has no reference at all to decision boundaries or neural networks -- he generally measures curvature of functions on ND inputs. This is not to say that contribution #1 is insignificant. You note that previous studies have used the Hessian alone, which you identify as incorrect. It appears to me that your use of the normalized Hessian *for this application* is new, and you present a clear (possibly new) argument for why one would need to normalize by the Gradient. I think a nod to the mathematical literature that has suggested this solution for some time (it is also explained fairly clearly in a 1993 book by Frank Morgan, and probably in earlier work as well) would be sufficient. I don't need to see a response on this, I trust you to word the contribution appropriately.

---

### Author Response · Authors · 2022-08-02
**Response to all Reviewers**

We thank the reviewers for their thorough and detailed reviews.

In response to reviewer comments requesting additional experiments, we present two additional experiments - the first on robustbench evaluation, and the second an ablation study. We describe these in detail below. In addition to this comment, we also provide detailed responses to each reviewer individually.

---

**Robustness evaluation on Robust Bench / AutoAttack**

As requested by reviewers **b1TK** and **o9LW**, we present robustness evaluations from the robustbench / auto-attack library (https://robustbench.github.io/), which consists of an ensemble of four parameter-free attacks. These include two variants of AutoPGD, a step-size free variant of the canonical PGD attack, and these are termed "APGD-CE" and "APGD-t" respectively. Other attacks include “FAB-t” which also automatically finds the appropriate perturbation norm, and “Square” which is a black-box attack. In all cases, our threat model is $\ell_2$ perturbations. In the tables below, we use an $\epsilon=0.1$ for relevant attacks, and we present results on a restricted test set of only 1000 test examples due to the lack of time in the rebuttal period. We plan to update the full results subsequently.

Our results below confirm the trend observed for the $\ell_2$ PGD case - adversarial training does the best closely followed by LCNN+GradReg. This should not be surprising, as LCNNs provably have smooth gradients (section 4.1) which ensures the absence of gradient masking, making such results transfer.

|  Model            | Auto-attack Acc. | Clean Acc. |
| ----------------- | ---------------- | ---------- |
| Standard          | 23.99%           | 77.62%     |
| LCNN              | 24.29%           | 76.92%     |
| GradReg           | 50.50%           | 77.52%     |
| LCNN+GradReg      | 51.51%           | 77.42%     |
| CURE              | 48.19%           | 75.71%     |
| Softplus+wt decay | 24.80%           | 77.42%     |
| Adv Training      | 56.25%           | 77.22%     |


  ---
**Ablation studies to evaluate effects of proposed changes**

As requested by reviewers **b1TK** and **o9LW**, we present ablation studies where we train models with each of the proposed modifications separately, i.e., we train a model with only spectral norm for the conv layers, gamma-Lipschitz Batchnorm or centered softplus. Our results show that for the resnet18 architecture considered, (1) performing spectral normalization had no effect of curvature as presumably the batchnorm layers are able to compensate for the lost flexibility, and (2) either penalizing the batchnorm alone or the softplus alone performs almost as well as LCNN, or sometimes even better in terms of curvature reduction.

However note that while in practice for Resnet-18 only $\gamma$-Lipschitz or centered softplus is sufficient for curvature reduction, in theory we require to regularize all components to avoid scaling restrictions in one layer being compensated by other layers, as dictated by the upper bound. In particular, this means that the strategy of penalizing only a subset of layers may not generalize to other architectures.

| Model     | Avg. Curv | Avg. Hess | Avg. Grad | Accuracy |
| --             | --        | -- | --         |     --    |
| ConvSpectralNorm only  | 358.86    | 8380.55      | 23.92     | 77.55%   |
| $\gamma$-BN only       | 65.78     | 1086.95      | 20.86     | 77.33%   |
| c-Softplus only | 57.49     | 734.05       | 16.99     | 77.31%   |
LCNN | 69.40 | 1143.62 | 22.04 | 77.30% |

---

In addition to this, we also respond individually to each reviewer to address their respective concerns.

Thank you!

---

> ### Author Response · Authors · 2022-08-06
> **Additional Robustness Results on Additional Baselines**
>
> **Additional robustness results and additional baselines**:
>
> As a follow-up on the robustness results presented above, as requested by reviewers **b1TK** and **o9LW**, we present here additional robustness results to disentangle the effect of the different constituent attacks within Auto-attack. In particular, we present results on targeted AutoPGD (APGD-t) and Square, which is a black-box attack. We evaluate these on the entire CIFAR100 test set, as opposed to the partial evaluation done in the comment above. These results are also broadly in line with our expectations, and the results presented in the paper, i.e., adversarial training performs the best closely followed by our LCNN+GradReg, however Adversarial training loses clean accuracy in the process.
>
> In addition, we also present results for additional baselines, namely, CURE + GradReg, as requested by reviewer **1uxf** and LCNN + GradReg + Adversarial training, as requested by reviewer **o9LW**. Similar to adversarial training, these baselines sacrifice clean accuracy for robust accuracy.
>
>
> |                               | APGD-t Acc. | Square Acc. | Clean Acc. |
> | ----------------------------- | ----------- | ----------- | ---------- |
> | Standard                      | 22.30%      | 53.30%      | 77.34%     |
> | LCNN                          | 23.90%      | 52.60%      | 77.22%     |
> | GradReg                       | 50.70%      | 65.20%      | 77.01%     |
> | LCNN+GradReg                  | 52.90%      | 65.20%      | 77.24%     |
> |||||
> | CURE                          | 50.50%      | 64.00%         | 76.54%     |
> | Softplus+wt decay             | 24.10%      | 54.20%      | 77.24%     |
> | Adv Training                  | 55.60%      | 67.40%      | 76.13%     |
> |||||
> | CURE + GradReg                | 61.30%      | 67.90%      | 74.79%     |
> | LCNN + GradReg + Adv Training | 59.70%      | 67.40%      | 75.99%     |

---

### Meta-Review · Area_Chair_vaYm · 2022-08-30

**Recommendation:** Accept
**Confidence:** Less certain

**Metareview:**

This paper studies low-curvature training of neural networks. Authors first propose a normalized curvature metric (that is invariant to the scaling of the gradient) and provide an upper bound for it in terms of the curvatures and slopes of different layers. Then they move on to the main contribution which is introducing an alternative activation and batch-normalization components which effectively limits the curvature. Finally, they show that when trained with these alternative components, one can achieve the same accuracy while having more stable gradients and more robust networks without significant training time increase. In my opinion, the fact that the robustness to input perturbations and the stable gradients come with simply substituting the previous components with the new one is the biggest advantage. On top of that, in terms of the training time, the proposed method is superior to other robust baseline approaches.

While I'm recommending acceptance, I strongly suggest authors to make the following improvements for the camera-ready to increase the impact of their work significantly:

1) More data & more models: Current experiments are very limited in terms of architectures and datasets. I suggest adding ImageNet results and two other small datasets (say SVHN and Fashion MNIST). Also, I suggest repeating the experiments on some none-ResNet architectures as well.

2) Authors have argued that the stable gradients are useful for interpretably. While this might have been established separately, it is still interesting to have at least one experiment to demonstrate this.

3) Even though this method is faster than other robust baselines, it is still 1.6x slower than standard training which is a significant limitation for adaptation of these components. I don't see an inherent reason for this slowness. Would be great if the implementation can be further optimized in terms of the running time.

4) One thing that I think is missing at the moment is a clear bottom line written somewhere regarding what we can conclude from the experiments (all this information is in the paper, but I think it would be better if it was highlighted somehow as “main conclusions” or sth of that form).


**Award:**

No

---

### Decision · Program_Chairs · 2022-09-14

Accept